# Redefining Autoimmune Disorders’ Pathoetiology: Implications for Mood and Psychotic Disorders’ Association with Neurodegenerative and Classical Autoimmune Disorders

**DOI:** 10.3390/cells12091237

**Published:** 2023-04-25

**Authors:** George Anderson, Abbas F. Almulla, Russel J. Reiter, Michael Maes

**Affiliations:** 1CRC Scotland & London, Eccleston Square, London SW1V 1PG, UK; 2Department of Psychiatry, Faculty of Medicine, Chulalongkorn University, Bangkok 10330, Thailand; 3Medical Laboratory Technology Department, College of Medical Technology, The Islamic University, Najaf 54001, Iraq; 4Department of Cell Systems and Anatomy, UT Health Long School of Medicine, San Antonio, TX 78229, USA

**Keywords:** autoimmunity, mitochondria, melatonin, N-acetylserotonin, aryl hydrocarbon receptor, TrkB, gut microbiome, circadian, cancer, treatment

## Abstract

Although previously restricted to a limited number of medical conditions, there is a growing appreciation that ‘autoimmune’ (or immune-mediated) processes are important aspects of a wide array of diverse medical conditions, including cancers, neurodegenerative diseases and psychiatric disorders. All of these classes of medical conditions are associated with alterations in mitochondrial function across an array of diverse cell types. Accumulating data indicate the presence of the mitochondrial melatonergic pathway in possibly all body cells, with important consequences for pathways crucial in driving CD8^+^ T cell and B-cell ‘autoimmune’-linked processes. Melatonin suppression coupled with the upregulation of oxidative stress suppress PTEN-induced kinase 1 (PINK1)/parkin-driven mitophagy, raising the levels of the major histocompatibility complex (MHC)-1, which underpins the chemoattraction of CD8^+^ T cells and the activation of antibody-producing B-cells. Many factors and processes closely associated with autoimmunity, including gut microbiome/permeability, circadian rhythms, aging, the aryl hydrocarbon receptor, brain-derived neurotrophic factor (BDNF) and its receptor tyrosine receptor kinase B (TrkB) all interact with the mitochondrial melatonergic pathway. A number of future research directions and novel treatment implications are indicated for this wide collection of poorly conceptualized and treated medical presentations. It is proposed that the etiology of many ‘autoimmune’/‘immune-mediated’ disorders should be conceptualized as significantly determined by mitochondrial dysregulation, with alterations in the mitochondrial melatonergic pathway being an important aspect of these pathoetiologies.

## 1. Introduction

Most, if not all, medical conditions are associated with alterations in mitochondrial and metabolic function, including cancer [1], dementias [2], cardiovascular diseases [3], psychiatric disorders [4,5], pulmonary diseases [6] and metabolic disorders [7]. Typically, the mitochondrial dysfunction involves the suppression of mitochondrial energy production from oxidative phosphorylation (OXPHOS), coupled with increased reactive oxygen species (ROS) and the involvement of reactive cells, either systemic immune cells and/or CNS glial cells [8]. The mitochondrial dysfunction is not generally limited to cells or tissues classically thought to underpin a given medical condition, but it may also be present in other organs and tissues or in circulating cells, such as platelets [9]. This poses problems for modern Western medicine, which has developed specialties derived from improving technologies that encourage a more detailed ‘atomistic’ approach to medical conditions. More ‘holistic’ conceptualizations of human medical conditions have been given considerable impetus from the data showing the gut microbiome/permeability to have strong pathoetiological and pathophysiological links to the above general medical categories. Wider systemic processes have also been shown to be ubiquitously relevant across diverse medical conditions, including changes in the patterned immune response [10] and circadian regulation [11]. Over a decade ago, it was treated with incredulity that Parkinson’s disease may have its origins in the gut, although this was first indicated by James Parkinson in 1817 in his classical description of the ‘shaking palsy’, to which he now lends his name [12].

The interface of ‘atomistic’ and ‘holistic’ data has always posed problems for medical classification and derived treatment, where a preference for simple biomarkers for a given cell or tissue is preferred due to its provision of a ready treatment target. This has proved problematic for more complex conditions, such as Alzheimer’s disease, where the dogged persistence in pathophysiologically defining Alzheimer’s disease by increased amyloid-β plaques and hyperphosphorylated tau-linked tangles for over 40 years has led to very little treatment improvement [13]. Recent data on other poorly conceptualized conditions, amyotrophic lateral sclerosis (ALS) and type 1 diabetes (T1DM), indicate that targeting the gut microbiome can slow/stop progression [14,15,16,17]. Decades of work focusing on the detailed alterations in motor neurons in ALS and pancreatic β-cells in T1DM have provided little clinical benefit.

The investigation of mitochondrial function across diverse medical conditions has very much come from an ‘atomistic’ perspective. However, recent data are showing how more systemic processes can regulate mitochondrial function, thereby providing a transformation in medical classification, treatment and prevention. This article reviews mitochondrial data across diverse medical conditions, linking this to wider systemic processes, and highlighting the role of the mitochondrial melatonergic pathway as a significant site for local and systemic-driven changes in intercellular homeostasis that underpin many medical conditions. The suppressed capacity of a cell to maintain an optimized mitochondrial function, including via the suppression of the tryptophan-melatonin pathway, underlies the growing number of medical conditions labelled as ‘autoimmune’ or ‘immune-mediated’ diseases.

This article reviews how gut and circadian processes regulate and reinforce homeostatic intercellular interactions, partly mediated via the capacity of these systemic processes to regulate the mitochondrial melatonergic pathway. Local intercellular processes that act to limit the capacity of a cell in a given tissue/microenvironment to upregulate the tryptophan-melatonin pathway trigger mitochondrial processes that underpin cell loss in the course of what is classically termed ‘autoimmune’ processes. These pathophysiological processes underpin the association of mood disorders and wider psychiatric disorders with a heightened susceptibility to ‘autoimmune’/‘immune-mediated’ conditions.

## 2. Classical ‘Autoimmunity’ and Mitochondria

### 2.1. Classical Autoimmunity

Autoimmunity is classically defined as the presence of T cells or antibodies that react to self-proteins within healthy cells. To some extent this is evident in all people, but it is classified as an autoimmune disease when this self-reactivity leads to tissue damage (either systemic or local). There are numerous well-established ‘autoimmune’ (or immune-mediated) diseases, such as multiple sclerosis, type 1 diabetes (T1DM) and rheumatoid arthritis as well as a growing number of diseases that have an important autoimmune aspect to their pathophysiology, such as Parkinson’s disease [18], amyotrophic lateral sclerosis (ALS) [19] and Alzheimer’s disease [20], with developing treatments now being shaped by the ‘autoimmune aspects’ of these classical neurodegenerative conditions [21].

A number of theories have been proposed in an attempt to explain the pathoetiology of classical autoimmune diseases, focusing on alterations in the B-cell antibody and cytokine production [22] and dysregulated suppressor cells, such as regulatory T cells (Treg) [23] and myeloid-derived suppressor cells (MDSC) [24]. Genetic risk factors predominantly include variants in MHC, immunoglobulins and T cell receptors, although not all susceptibility genes lead to an individual having an autoimmune disorder. Many susceptibility genes seem associated with the risk for distinct medical conditions, whilst others, such as single nucleotide polymorphisms (SNP) in protein tyrosine phosphatase non-receptor type 22 (PTPN22), are relevant to a growing list of autoimmune disorders, including T1DM and rheumatoid arthritis [25]. This indicates general and specific risk factors.

Epigenetic processes are a crucial aspect of ‘autoimmunity’ in many of its manifestations [26], including the following: (i) histone acetyltransferase (HAT) and histone deacetylase (HDAC), which regulate histone acetylation and alter histone spatial structure, thereby impacting on transcription. The gut microbiome influence on ‘autoimmunity’ includes variations in the levels of gut microbiome-derived butyrate, an HDAC inhibitor (HDACi), and regulator of the immune response in a number of ‘autoimmune’ conditions [27], including via enhanced Treg differentiation [28]; (ii) miRNAs, long non-coding RNAs (lncRNAs) and circular RNAs (CircRNA). miRNAs and the RNA-induced silencing complex (RISC) suppress mRNA transcription via binding to the mRNA 3′UTR region, including via complex formation. This has relevance to autoimmune demyelination [29], as well as a general impact on patterned gene expression. CircRNA can sponge miRNAs to regulate mRNA transcription, as shown in rheumatoid arthritis and systemic lupus erythematosus (SLE) [30]. LncRNA may also sponge miRNA and regulate DNA transcription, with relevance in rheumatoid arthritis [31]; (iii) DNA methyltransferase (DNMT) and ten eleven translocation (TET) modulate gene transcription via the hypermethylation and hypomethylation of cytosine in DNA, respectively. DNMT-1 is proposed to regulate immune responses, as shown in Graves’ disease [32], whilst Tet methylcytosine dioxygenase 2 (Tet2) deficiency drives hepatic microbiome dysbiosis, leading to CD8^+^ T cell-mediated autoimmune hepatitis [33]; (iv) N^6^-methyladenosine (m^6^A) is a common modification of both mRNA and DNA, whereby the METTL3/METTL14 methyltransferase complex installs m^6^A onto mRNA, which can impact not only translation, but also nuclear export and decay, as well as pre-mRNA processing, with relevance to immune regulation in T1DM [34].

### 2.2. Mitochondrial Metabolism and ‘Autoimmunity’

Epigenetic processes also regulate mitochondrial function. The m^6^A methyltransferase METTL3 leads to mitochondrial dysfunction via mRNA modification of the ‘master mitochondrial regulator’, peroxisome proliferator-activated receptor gamma coactivator (PGC)-1α [35]. HDAC regulates a number of autoimmune disorders, with its inhibition by butyrate optimizing mitochondrial function [27]. Autoimmunity-associated suboptimal mitochondrial function can drive oxidative and nitrosative stress (O&NS) that induces miRNAs and lncRNAs, shown to be significant aspects of SLE pathophysiology [36], whilst a number of miRNAs increased in T1DM are strongly associated with suboptimal mitochondrial function [37]. DNMT-1 hypermethylation contributes to suboptimal mitochondrial physiology in diabetes-linked retinopathy [38].

For the four epigenetic processes highlighted above, alterations in mitochondrial function may be an integral aspect of both classical autoimmune disorders and ‘immune-mediated’ disorders, such as Parkinson’s disease, schizophrenia and ALS. Recent data showed intestinal Gram-negative bacteria infection to trigger Parkinson’s disease-like symptoms in PINK1 ko mice, which can be successfully treated with L-DOPA [39]. This is parsimonious with a gut-linked early pathophysiology of Parkinson’s disease [12,40] as well as with mutations in the PINK1 and PRKN ubiquitin ligase genes as risk factors for an early-onset form of Parkinson’s disease [39]. PINK1 and parkin regulate adaptive immunity via the repression of mitochondrial antigens, indicative of classical autoimmune-like processes that can induce mitochondria-specific cytotoxic CD8^+^ T cells systemically and in the CNS. Such data are compatible with PINK1 as an immune system repressor, with intestinal infection as a triggering event in Parkinson’s disease and possibly many other currently poorly conceptualized and treated diseases. This is supported by data showing PINK1 to suppress the presentation of mitochondrial peptides on MHC-1 to the immune system following LPS infection [41]. The presentation of mitochondrial antigens on MHC-1 is followed by the formation of ’anti-mitochondrial’ CD8^+^ T cells [42]. The authors modeled the effects as arising from gut mitochondria antigens priming the CD8^+^ T cell response, with subsequent infiltration into the CNS [42], linking to data showing inflammatory bowel disease to be a Parkinson’s disease risk factor [43]. Such data highlight how alterations in the gut microbiome can modulate mitochondrial function and ‘autoimmune-linked’ processes.

As well as the gut-mediated antigen priming of CD8^+^ T cells that subsequently enter other organs, the gut can have ‘bystander’ activation effects on CD8^+^ T cells, as recently shown in T1DM [44]. These authors showed that the initial activation of pancreatic islet-specific CD8^+^ T cells takes place in the pancreatic lymph nodes, with additional bystander enhancement of effector functions occurring in the gut’s Peyer’s patches [44]. Interestingly, this study also showed orally administered butyrate in preclinical T1DM models to suppress this ‘bystander’ potentiation of CD8^+^ T cell autoreactive effector functions [44], highlighting how the pre-existing state of the gut microbiome may be a relevant factor in ‘autoimmunity’ and symptom severity. Pancreatic β-cells also show suppressed PINK1/parkin-mediated mitophagy during T1DM development arising from an increase in oxidative stress [45], paralleling similar effects shown in substantia nigra pars compacta dopamine neurons [46]. The suppression of mitophagy allows dysfunctional mitochondria to be maintained, thereby dysregulating energy and oxidant production (see Figure 1).

Notably, the most commonly used preclinical T1DM model, streptozotocin treatment, suppresses the mitochondrial melatonergic pathway, as shown in the retina [47], which is highly suggestive of the importance of the mitochondrial melatonergic pathway in T1DM pathoetiology and pathophysiology [17]. As melatonin upregulates PINK1 and mitophagy [48], such data indicate that mitochondrial melatonergic pathway suppression may be an important driver of immune-mediated cell loss and autoimmune responses across diverse medical conditions. As well as streptozotocin suppression of the melatonergic pathway in T1DM models, data suggest that the melatonergic pathway is suppressed in the substantia nigra pars compacta and other CNS areas affected in Parkinson’s disease patients [49]. Such data are indicative of a significant role for the mitochondrial melatonergic pathway in ‘autoimmune’-linked processes as indicated by oxidative stress suppression of PINK1 and mitophagy that upregulates MHC-1 in association with the attraction of CD8^+^ T cells in the course of ‘autoimmune’ responses [46]. An important consequence of suppressed melatonin, PINK1 and mitophagy is ROS upregulation arising from suboptimal and dysfunctional mitochondria, leading not only to oxidative stress but to an increase in ROS-driven miRNAs, thereby impacting patterned gene induction, with consequences not only for the cell’s survival but for communication with neighboring cells, including immune effector cells such as natural killer (NK) cells and CD8^+^ T cells.

Many ‘immune-mediated’ conditions show PINK1 alterations linked to suboptimal mitochondrial function, suggesting that ‘immune-mediated’ conditions may be primarily metabolic disorders, many of which will be modulated, if not driven, by the dysregulation of mitochondria and the mitochondrial melatonergic pathway. The pro-inflammatory cytokine, interleukin (IL)-1β-induced specific protein (SP)1, increases miR-144-3p to suppress PINK1/parkin-mediated autophagy in a rheumatoid arthritis model [50]. As the ROS-induced NLR family pyrin domain containing 3 (NLRP3) inflammasome is typically upstream of IL-1β (and IL-18) induction, the ROS/nuclear factor kappa-light-chain-enhancer of activated B-cells (NF-kB)/NLRP3/IL-1β [51] suppression of PINK1/parkin and subsequent mitophagy suppression may be intimately linked to ROS-driven miRNAs and patterned gene expression, and therefore to the initial mitochondrial oxidant/antioxidant ratio status. The inability to counter the oxidant challenge is not incompatible with suppression of the mitochondrial melatonergic pathway in the course of immune-mediated cell loss/damage. Another commonly investigated pro-inflammatory cytokine, tumor necrosis factor (TNF)α, also inhibits PINK1-mediated mitophagy, leading to altered mitochondrial function and enhanced cytosolic mtDNA levels. This has a number of consequences that include the cyclic guanosine monophosphate-AMP synthase (cGAS)-stimulator of interferon genes (STING) pathway induction, with mtDNA binding to cGAS following TNFα [52]. Such data provide ready association of oxidative stress and pro-inflammatory cytokines to the regulation of mitochondrial function in driving the pathophysiological underpinnings of ‘autoimmune’ disorders. Overall, the capacity of mitochondria to regulate their oxidant/antioxidant ratio status, including via the induction of the mitochondrial melatonergic pathway, is an important determinant of the association of ‘autoimmunity’ with PINK1 suppression, pro-inflammatory cytokines and mitophagy as well as patterned miRNAs and gene expression (see Figure 2).

Raised cerebrospinal fluid and plasma indicants of autophagy and mitophagy, including PINK1 and parkin, are evident in the active phase of relapse remitting multiple sclerosis, with autophagy inhibition enhancing myelination, highlighting that the enhancement, as well as the suppression, of mitophagy are aspects of ‘autoimmune’ disorders [53]. Dysregulated mitophagy is evident in many classical ‘autoimmune’ disorders, including rheumatoid arthritis [54], SLE [55], T1DM [56], myasthenia gravis [57] and autoimmune hepatitis [58], as well as in suspected ‘autoimmune’ conditions, such as endometriosis [59], and neurodegenerative conditions [60]. Notably, alterations in mitochondrial function and mitophagy are not always evident in cells classically associated with these conditions, but in cells relevant to wider intercellular interactions, including immune cell mitochondria, as evident in Treg in myasthenia gravis [57]. Such data implicate a ‘microenvironment’ of diverse interacting cells with their mitochondrial function dynamically shaped via alterations in the mitochondrial function of other ‘microenvironment’ cells, paralleling the dynamic intercellular interactions occurring within the tumor microenvironment [61], and as recently proposed for T1DM [17].

The dynamic metabolic intercellular interactions in a given microenvironment, including tumors [62], are problematic to detail given the relative paucity of studies investigating intercellular mitochondrial interactions, and the role of the mitochondrial melatonergic pathway to the mitochondrial interactions within such microenvironments. Apart from macrophages and microglia [63,64], it is unknown as to whether other immune cells express the mitochondrial melatonergic pathway and whether the regulation of this pathway is relevant to patterned immune responses across immune-mediated, ‘autoimmune’ medical conditions. The relative paucity of data on the mitochondrial function across immune cells is a historical legacy of classifying and subtyping immune cells on the basis of plasma membrane proteins. Given the plethora of data showing exogenous melatonin and pineal/circadian melatonin to significantly regulate immune cells, it would seem highly likely that variations in the regulation of the mitochondrial melatonergic pathway across immune cells would be a fruitful area of investigation. The role of the mitochondrial melatonergic pathway is covered in Section 3.1 below.

An autoimmune disease that is often, but not exclusively, seen in SLE patients is antiphospholipid antibody syndrome (APAS), which leads to a raised risk of thrombosis, in association with heightened oxidative stress [65]. APAS associates with symptom severity and fatality following severe acute respiratory syndrome-coronavirus 2 (SARS-CoV-2) infection [66], as well as occasionally with SARS-CoV-2 vaccines [67]. Thrombocytopenia and alterations in platelet function are evident in APAS [68], indicating a role for platelets in this, and other, autoimmune disorders [69]. Platelets can store IL-1β in their cytosol, as well as releasing platelet microparticles (PMPs) [70], and mitochondria [71], all of which can influence the mitochondrial function and activity of many cells, including immune cells [72,73]. Recent data indicate that PMPs activated by anti-β_2_GPI/β_2_GPI complexes upregulate the NLRP3 inflammasome in PMPs, with the NLRP3 inflammasome being the major driver of endothelial cell pyroptosis in the course of APAS [74]. Platelets are also a major source of circulating amyloid-β and α-synuclein in Alzheimer’s disease and Parkinson’s disease, respectively [75], indicating a platelet role in the classical pathophysiology of these neurodegenerative conditions, including their association with ‘autoimmune’ processes. However, an overlooked aspect of platelet function is their uptake and storage of serotonin, especially in the provision of serotonin as a precursor for the mitochondrial melatonergic pathway across different ‘autoimmune’ disorders. The regulation of serotonin and the major melatonergic pathway products, NAS and melatonin, is an overlooked aspect of platelet function and pathophysiological relevance, including within the tumor microenvironment [61], and multiple sclerosis [8] (see Figure 3).

Overall, the above highlights how alterations in mitochondrial function are integral to local cell/tissue loss in ‘autoimmune’ disorders, as well as triggering immune-mediated cell damage/loss. It also highlights that alterations in mitochondrial function in immune cells, including NK cells, but especially CD8^+^ T cells, are also important, as has been long recognized from the induction of ‘exhaustion’ in NK cells and CD8^+^ T cells in the tumor microenvironment. The role of the gut microbiome in a wide array of ‘autoimmune’ disorders is intimately linked to the capacity of the gut microbiome/permeability to regulate mitochondrial function in damaged/lost cells in ‘autoimmune’ disorders as well as in the regulation of the immune response in a given ‘autoimmune’ disorder microenvironment. The gut also has important, more direct, impacts on mitochondrial function in immune cells via the antigen priming of CD8^+^ T cells or ‘bystander’ enhancement of the CD8^+^ T cell effector function [42,43].

The above highlights how alterations in mitochondrial function are an integral aspect of how cytolytic cells associate with ‘autoimmune’ disorders. However, another important aspect of autoimmunity is the production of antibodies by B-cells.

### 2.3. B-Cells, Antibodies and Mitochondria

The process of activation in stimulated naïve B-cells involves OXPHOS and tricarboxylic acid (TCA) cycle upregulation as well as nucleotide biosynthesis but does not involve the upregulation of glycolysis, unlike most immune cells in the process of activation [76]. The suppression of OXPHOS or glutamine uptake markedly impairs B-cell growth and differentiation, limiting their capacity to produce antibodies and cytokines, as well as suppressing their capacity to present antigens to T cells [77].

Upon encountering an antigen, naïve B-cells differentiate into antigen-specific memory B-cells or antibody-secreting plasma cells, affording longer-term protection against any subsequent antigen exposure. In contrast to long-lived plasma cell residence in the bone marrow, memory B-cells continue to circulate, being able to rapidly differentiate upon re-encountering their specific antigen. As popularly highlighted by the COVID-19 pandemic, memory B-cell longevity is variable, with the processes underpinning that variability still unclear [78]. Emerging data indicate that mitochondrial function and mitophagy are important processes regulating memory B-cell survival and function, which may be at least partly dependent upon the classically defined energy sensor, adenosine monophosphate–activated protein kinase (AMPK), which acts to increase ATP production, autophagy, fatty acid oxidation and mitochondrial biogenesis whilst concurrently suppressing protein and fatty acid synthesis [79]. Aging-linked deficits in B-cell mitochondrial function may also be attenuated by AMPK, [80], highlighting the relevance of mitochondrial regulation in B-cell function and antibody response over aging. In comparison with other lymphocytes, B-cells show a relatively high mitochondrial metabolism, heightened mitophagy and ROS, as well as higher levels of mitochondrial membrane potential and mitochondrial mass, especially in pro B-cells [81]. Mitophagy in B-cells is driven by PINK1/parkin2 [81]. The knockdown of the cardiolipin transacylase enzyme, tafazzin, also attenuates mitochondrial function in B-cells, leading to a reduction in antibody production [82], and further highlighting the functional importance of the mitochondrial metabolism in B-cells.

As in all cells, microRNAs are important determinants of B-cell function and patterned responses [83]. As many miRNAs are ROS-regulated, alterations in mitochondrial function and mitochondrial ROS production will drive variations in miRNAs, thereby shaping patterned gene expressions in B-cells. This is an important route for mitochondrial function to modulate patterned gene expression and thereby variations in cell function and, occasionally, cell phenotype. The presence and regulation of the mitochondrial melatonergic pathway in B-cells are therefore likely to be of some importance to the regulation of patterned gene expression, as is the induction of miRNAs that regulate the melatonergic pathway, including miR-7, miR-375, miR-451 and miR-709 [84,85].

As well as melatonin, the immediate precursor of melatonin, N-acetylserotonin (NAS), may be of relevance to B-cell regulation. NAS is a brain-derived neurotrophic factor (BDNF) mimic via its activation of TrkB [86]. The BDNF activation of TrkB contributes to B-cell maturation in the bone marrow, as well as promoting B-cell differentiation in the course of B-cell malignancies [87]. The truncated TrkB-T1 (TrkB95) is expressed on activated B-cells as well as being important to B-cell maturation in the bone marrow [88]. Activated B-cells can also secrete BDNF [89]. As the melatonin precursor, NAS, activates TrkB [86], such data would indicate a role for variations in the NAS/melatonin ratio in the maturation and function of B-cells, thereby more directly linking B-cell development and antibody production to variations in mitochondrial function, both in B-cells and B-cell interacting cells, via autocrine and paracrine NAS, respectively.

Notably, melatonin inhibits apoptosis of B-cells during development and when challenged, via suppressing oxidative stress and NF-kB [90,91]. Aging-linked deficits in B-cell function are intimately linked to alterations in mitochondrial function, including oxidative stress-induced mitochondrial DNA mutations [92]. It clearly requires investigation as to the presence and expression of the mitochondrial melatonergic pathway in B-cells. The potential importance of the melatonergic pathway in B-cells is highlighted by the data showing the transcription factor yin yang (YY)1 to be crucial in the regulation of B-cell mitochondria gene expression as well as the immunoglobulin (Ig) class switch recombination (CSR) that occurs in activated B-cells [93]. CSR changes the Ig constant region to produce the best Ig isotype to protect against a given pathogen, such as shifting from isotype IgM to IgD. Notably, YY1, as with NF-kB, can upregulate the melatonergic pathway in other cell types [94], suggesting that this important function of YY1 in the regulation of B-cell mitochondrial mass, membrane potential and CSR [94,95] may be linked to the regulation of the mitochondrial melatonergic pathway in B-cells. As well as the relevance to suppressed antibody responses to common viral infections, such as coronavirus and influenza, the regulation of mitochondrial function in B-cells is important to a wide array of diverse medical conditions now conceptualized as ‘immune-mediated’, as well as to the general deficits in immune response over the course of aging, which underpin many fatalities across diverse medical conditions (see Figure 4).

B-cells that act to suppress immune responses are collectively referred to as regulatory B-cells (Bregs). Bregs are generally regarded as helping to maintain tolerance and re-establishing homeostasis, primarily via the release of IL-10, IL-35 and transforming growth factor (TGF)-β, as well as via the expression of the immune checkpoint inhibitor ligand, programmed cell-death ligand 1 (PD-L1) [95]. Recent work has highlighted a significant role for Bregs in ‘autoimmunity’, cancers, infection, allergy and metabolic disorders [95]. Peroxisome proliferator-activated receptor gamma (PPAR-γ)-deficient B-cells show suppressed IL-10 production and suppressive ability, with PPAR-γ agonists significantly expanding IL-10 producing B-cells [96]. The knockdown of cardiolipin transacylase enzyme, tafazzin, in mesenchymal stem cells (MSC) suppresses MSC mitochondrial function, inducing a Breg phenotype in LPS-activated B-cells [97]. Such data highlight the importance of variations in intercellular interactions and the intercellular consequences arising from alterations in mitochondrial function in a given cell within an intercellular microenvironment.

As with many other medical conditions, ‘autoimmune’ disorders seem invariably to be associated with an increase in pro-inflammatory transcription factors, including NF-kB [98] and YY1 [99]. This may be an important commonality as both NF-kB and YY1 can upregulate the mitochondrial melatonergic pathway, which may be crucial in determining the course of activation–deactivation in immune cells [63,64]. NF-kB and YY1 are important regulators of both aspects of classical ‘autoimmunity’, namely antibody production by B-cells and autoreactivity in cytolytic T cells, with consequences for the mitochondrial function in both aspects of autoimmunity. As noted, the gut microbiome/permeability is a significant regulator of ‘autoimmune’ responses, and it is covered in the next section.

## 3. Gut Microbiome and Systemic Mitochondria

Alterations in the gut microbiome and gut permeability are cutting-edge areas of research across a wide array of diverse medical conditions. This arises from the impact that gut microbiome-derived products can have on mitochondrial function [100]. The most explored gut microbiome-derived products are the short-chain fatty acids, propionate, acetate and, especially, butyrate. Butyrate is a histone deacetylase inhibitor (HDACi) and therefore an epigenetic regulator. Butyrate can also mediate effects via the G-protein coupled receptors (GPR)-41 and GPR-43 [101]. Butyrate predominantly optimizes the mitochondrial function by upregulating the mitochondria-located sirtuin-3, which deacetylates and disinhibits the pyruvate dehydrogenase complex (PDC), thereby driving an increased conversion of pyruvate to acetyl-CoA. Acetyl-CoA is necessary to increase ATP production by the tricarboxylic acid (TCA) cycle and by OXPHOS. This is an important means whereby gut microbiome-derived butyrate can optimize mitochondrial function across systemic and CNS cells.

Butyrate-induced acetyl-CoA is a necessary co-substrate for the initiation of the mitochondrial melatonergic pathway, whereby serotonin is converted to N-acetylserotonin (NAS) by the enzymatic action of aralkylamine N-acetyltransferase (AANAT) in the presence of acetyl-CoA. NAS is then converted to melatonin by acetylserotonin methyltransferase (ASMT). The capacity of butyrate to upregulate the mitochondrial melatonergic pathway, along with the induction of ATP from the TCA cycle and OXPHOS, allows variations in the products of the gut microbiome to have significant and sometimes dramatic effects on a host of body cells and body systems [17].

The gut microbiome is also important in tryptophan regulation, which is a crucial substrate for serotonin production and therefore of serotonin as a necessary precursor for the initiation of the mitochondrial melatonergic pathway. Most tryptophan is diet-derived and taken up by gut cells before entering the circulation. However, the gut microbiome can also produce tryptophan (and the other aromatic amino acids, tyrosine and phenylalanine) via the shikimate pathway. The shikimate pathway is primarily achieved by *Akkermansia muciniphila* [102], allowing factors regulating *Akkermansia muciniphila*, such as bacteriophages and enteroviruses, to determine shikimate pathway activity and tryptophan, tyrosine and phenylalanine production. This has a number of implications, including for systemic tryptophan availability, tryptophan conversion to tryptamine that activates the aryl hydrocarbon receptor (AhR) to maintain the gut barrier and tryptophan availability for the tryptophan-melatonin pathway. The gut’s influence on tryptophan synthesis, uptake and availability coupled to butyrate production allows the gut and gut microbiome to powerfully regulate mitochondrial function across all tissues and organs, including in immune and glial cells, and thereby on the influence of these reactive cells in the pathoetiology and pathophysiology of numerous medical conditions [103].

### 3.1. Tryptophan and the Melatonergic Pathway

Tryptophan is important to mitochondrial regulation primarily via its capacity to be converted into melatonin, with the melatonergic pathway being an integral aspect of mitochondrial function and plasticity. Importantly, the mitochondrial melatonergic pathway seems ubiquitously expressed across all cells (so far investigated) in the three kingdoms of life on Earth, namely animals, plants and fungi. This is proposed to arise from the common ancestor of multicellular life on Earth emerging about 2 billion years ago when a melatonin-producing ancient bacteria crept into a single-cellular organism, with this melatonin-producing ancient bacteria eventually evolving into mitochondria [104]. The apparent maintenance of the mitochondrial melatonergic pathway across all subsequent forms of life on Earth indicates the importance of the mitochondrial melatonergic pathway to multicellular life. This seems to have occurred due to the powerful role that mitochondrial melatonin has as an antioxidant, anti-inflammatory, antinociceptive and optimizer of mitochondrial function. As interactions among bacteria and of bacteria with other challenges are invariably met by the bacterial release of oxidants, the capacity of these incorporated ancient bacteria to protect themselves against challenges seems to have been optimally met by melatonin. The uptake of tryptophan and its synthesis by the shikimate pathway in the gut are therefore intimately linked to optimized mitochondrial function across all body cells.

The melatonergic pathway also provides response plasticity to mitochondria, as exemplified by the differential effects of melatonin and its immediate precursor, NAS. Both NAS and melatonin have antioxidant effects. However, NAS is a brain-derived neurotrophic factor (BDNF) mimic via the NAS activation of the BDNF receptor, tyrosine receptor kinase B (TrkB) [86], allowing NAS to have trophic effects as well as BDNF-linked metabolic effects. Variations in the NAS/melatonin ratio can occur when the melatonergic pathway is induced, including from the pineal gland at night. Being a BDNF mimic allows NAS to have more proliferative effects than melatonin, with relevance to the induction of neurogenesis in the hippocampal dentate gyrus [105]. However, NAS production is problematic for conditions with excessive proliferation, such as cancers [106,107] and endometriosis [108,109].

Proliferative medical conditions such as cancers highlight another aspect of the mitochondrial melatonergic pathway across diverse medical conditions, namely how intercellular homeostatic interactions in a given microenvironment can be shaped by fluxes from one cell regulating the mitochondrial melatonergic pathway in another cell. Recent work indicates that cancer cells act to regulate the mitochondrial melatonergic pathway across different cells within the tumor microenvironment [61,62,110,111]. As such, the plasticity of the mitochondrial melatonergic pathway can be utilized by challenged/‘domineering’ cells (such as tumors) to determine the nature of the intercellular homeostatic interactions occurring within a given microenvironment. In tumors [112] and in pancreatic β-cells [113], this may be achieved by the pro-inflammatory cytokine induction of indoleamine 2,3-dioxygenase (IDO), which converts tryptophan to kynurenine. The IDO conversion of tryptophan to kynurenine not only deprives tryptophan for the serotonergic and melatonergic pathways, but also allows kynurenine to activate the aryl hydrocarbon receptor (AhR) and AhR-induced cytochrome P450 (CYP)1A2, which, along with ATP at the purinergic P2Y1r and glutamate at the metabotropic glutamate receptor 5 (mGluR5), can ‘backward’ convert melatonin to NAS via O-demethylation [114,115]. The AhR-induced cytochrome (CYP)1b1 can also contribute to the metabolism of melatonin, thereby being another means to increase the NAS/melatonin ratio. In tumors, NAS release activates the TrkB receptor to increase the survival and proliferation of cancer stem-like cells [116]. As tumors are driven by alterations in the metabolism, the tumor microenvironment can be conceived of as a network of evolutionary-derived bacteria (in the form of mitochondria) with the cells dynamically interacting to modulate their mitochondrial function, including via the regulation of the mitochondrial melatonergic pathway. This provides a markedly different conceptualization of intercellular interactions, which have classically focused on cellular releases acting on a plasma membrane receptor of another cell, leading to intracellular signaling pathways that increase transcription factors to induce patterned gene expression via effects in the cell nucleus. The regulation of the mitochondrial melatonergic pathway can dramatically alter such classical intercellular processes, including via alterations in mitochondrial reactive oxygen species (ROS), which drive ROS-dependent microRNAs (miRNAs) that then shape patterned gene responses and intercellular fluxes [117]. Overall, integrating the tryptophan-melatonin pathway as a core aspect of mitochondrial function into intercellular processes provides a perspective upon which changes in the gut microbiome may act (see Figure 5).

### 3.2. Gut Microbiome, Melatonergic Pathway and Mitochondria

Gut microbiome-derived butyrate does not simply act on a given cell to increase sirtuin-3/PDC/acetyl-CoA/TCA cycle/OXPHOS and the melatonergic pathway, but is also acting on, and being regulated by, intercellular processes. For example, the ‘backward’ conversion of melatonin to NAS limits the capacity of butyrate to increase melatonin production. The production and release of butyrate therefore have its effects partly determined by local intercellular processes that act on the mitochondrial melatonergic pathway. This is further complicated by the release of pro-inflammatory cytokines that accompany intercellular dysregulation (as in most medical conditions), which increase gut permeability and cause gut dysbiosis, thereby decreasing butyrate production. Suppressed butyrate levels are proposed to limit butyrate from prematurely inducing homeostasis, whilst the local resolution of the dynamic intercellular fluxes of local inflammation is being resolved. The gut microbiome/permeability can therefore be significantly regulated by variations in mitochondrial function and dynamic intercellular interactions across the body. Variations in the capacity to maintain the gut barrier can arise from the factors regulating the shikimate pathway (and tryptophan conversion to tryptamine to activate the AhR), and from local melatonin production by gut enterochromaffin cells as well as from the levels of butyrate produced [118]. An array of diverse factors, including genetic, dietary and stress factors influence the capacity of the gut to maintain the gut barrier, and therefore the influence that the gut has on systemic inflammatory processes. This has parallels to the effects of circadian melatonin, which dampens inflammation and optimizes/resets mitochondrial function, with pineal melatonin being suppressed following acute inflammation [119]. The gut may therefore have parallels with the immune–pineal axis proposed by Regina Markus and colleagues [63], which is covered in the next section.

The other widely investigated factor released by the gut following gut permeability is lipopolysaccharide (LPS), which activates toll-like receptor (TLR)4 in many body cells, typically triggering pro-inflammatory transcription factors such as the nuclear factor kappa-light-chain-enhancer of activated B-cells (NF-kB) and yin yang (YY)1. NF-kB and YY1 are strongly associated with the induction of inflammatory processes and reactivity in immune and glia cells. However, both transcription factors also induce the melatonergic pathway, thereby providing intracrine, autocrine and paracrine melatonin effects that time-limit the reactive responses of macrophages and microglia, with autocrine melatonin shifting these cells from an M1-like pro-inflammatory to an M2-like pro-phagocytic phenotype [63,64]. Factors dysregulating the mitochondrial melatonergic pathway in these immune cells, therefore, impact on the process of reactivation/deactivation. The suppressed capacity of astrocytes to upregulate mitochondrial melatonin production following TLR4 signaling by LPS (or endogenous ligands, such as high-mobility group box (HMGB)1 and heat shock protein (hsp)70) is proposed to lead to the maintained amyloid-β production in Alzheimer’s disease [13]. The gut permeability-induced LPS/TLR4/NF-kB/YY1 pathway is intimately associated with the capacity of mitochondria to upregulate the melatonergic pathway, including from gut permeability-associated LPS. This also indicates that intercellular mitochondrial interactions in a given tissue/microenvironment that suppress the melatonergic pathway in a given cell(s) will alter the impacts of gut permeability-derived LPS as well as butyrate.

The capacity to upregulate the melatonergic pathway is of considerable importance to mitochondrial function and cell survival. As well as having an antioxidant and anti-inflammatory effect, melatonin also induces endogenous antioxidants and antioxidant enzymes, such as glutathione (GSH) and catalase. Melatonin can also form a film over the mitochondrial outer membrane, thereby influencing membrane fluidity and the complexes formed by receptors and channels [120]. The suppression of the tryptophan-melatonin pathway, therefore, has an array of consequences for mitochondrial function, including ROS production and ROS-driven microRNAs (miRNAs) that can then shape patterned gene induction. This may have relevance to recent data showing tumor necrosis factor (TNF)α to remodel the outer mitochondrial membrane via the induction of the linear ubiquitin chain assembly complex (LUBAC), with this change in the mitochondrial outer membrane facilitating the transport of activated NF-κB to the nucleus [121]. Whether a melatonin film over the mitochondrial membrane from loosely binding to hundreds of different receptors occludes the capacity of LUBAC to remodel the mitochondrial membrane and enhance NF-kB-driven inflammatory transcriptions will be important to determine. As pro-inflammatory cytokines induce gut permeability and increase circulating LPS, the suppression of the mitochondrial melatonergic pathway may not only prolong, but also enhance, NF-kB-induced pro-inflammatory processes. This would be another mechanism whereby the mitochondrial melatonergic pathway interacts with gut permeability-driven circulating LPS, as well as endogenous TLR4 ligands, to heighten and prolong inflammatory transcription.

Overall, the two main gut microbiome-derived factors highlighted above, namely butyrate and LPS, mediate relevant effects across most medical conditions via impacts on mitochondrial function and dynamic intercellular interactions throughout the body. These effects are importantly modulated by variations in the capacity of a given cell to upregulate the mitochondrial melatonergic pathway, which may be determined by gut microbiome factors, such as the shikimate pathway and *Akkermansia muciniphila*.

## 4. Pineal Melatonin and Systemic Mitochondria

As with the gut microbiome, variations in circadian regulation are linked to a diverse array of medical conditions, including in the pathoetiology of cancers [122] and dementias [123], as well as many neuropsychiatric conditions [124,125]. Circadian dysregulation is strongly associated with the suppression of night-time melatonin production by the pineal gland, which decreases 10-fold between the ages of 18 years and 80 years [126]. As noted above, a number of factors can alter the NAS/melatonin ratio from the pineal gland, which shows alterations across a number of medical conditions [127], with consequences that have still to be determined.

Many of the seemingly ubiquitous benefits of exogenous melatonin are mediated via its capacity to upregulate mitochondrial function, in close association with melatonin’s capacity as an antioxidant, anti-inflammatory, antinociceptive and endogenous antioxidant inducer [128]. More recent work shows exogenous melatonin (circadian, autocrine, paracrine) to be taken up into mitochondria via the peptide transporters (PEPT)1/2, whilst the sulphation metabolites of melatonin can be taken up by the organic anion transporter (OAT)3 [129]. Exogenous melatonin can upregulate mitochondrial enzymes and antioxidant enzymes, including manganese superoxide dismutase (SOD)2 [130], which, similar to butyrate effects, is via the upregulation of sirtuin-3 [131]. Recent conceptualizations of pineal melatonin have highlighted its role at night in ‘resetting’ the mitochondrial function to OXPHOS, which dampens any ongoing inflammatory activity, especially as driven by immune and glial cells [132]. It is the loss of this resetting of mitochondrial function at night that is proposed to underpin cancer pathoetiology [132,133].

As well as via the induction of sirtuin-1 and sirtuin-3, pineal melatonin can disinhibit PDC and upregulate the conversion of pyruvate to acetyl-CoA via the circadian gene, brain and muscle ARNT-like 1 (BMAL1) [134,135]. The suppression of pineal melatonin over the course of aging, as well as by pro-inflammatory cytokines [119], LPS and the induction of some miRNAs, including miR-7 [136], curtails the capacity of pineal melatonin to upregulate ATP production by the TCA cycle and OXPHOS as well as limit pineal melatonin’s induction of the mitochondrial melatonergic pathway. This has a number of consequences, including modulating the effects of gut microbiome-derived butyrate in optimizing mitochondrial function, which seems at least partly dependent on the butyrate induction of the melatonergic pathway, as shown in intestinal epithelial cells [118].

The association of circadian disruption with many medical conditions, such as psychiatric disorders, Alzheimer’s disease and other aging-linked neurodegenerative conditions is intimately linked to suppressed pineal melatonin production [137], and the consequences that this has for mitochondrial function, especially the resetting of reactive cells, such as glia and immune cells. As with the gut microbiome, the pineal melatonin optimization of mitochondrial function (partly via the induction of the mitochondrial melatonergic pathway) better enables all cells to resist challenges posed by various stressors, including neighboring cells in a given intercellular microenvironment. There is a growing appreciation that a wide array of diverse medical conditions have ‘autoimmune’ aspects, such as Parkinson’s disease [138] and schizophrenia [139]. The suppression of the mitochondrial melatonergic pathway also has consequences for the regulation of processes underpinning immune-mediated cell loss, classically referred to as ‘autoimmunity’, including via the regulation of intercellular processes.

## 5. Local Intercellular Interactions, Mitochondrial Melatonin and Cell Elimination

Recent work has highlighted the importance of local intercellular interactions that may lead to a given cell type becoming dysfunctional, as in the interactions of pancreatic α-, δ- and ε-cells with pancreatic β-cells in driving pancreatic β-cell loss in T1DM [17]. As in the tumor microenvironment [61], such intercellular interactions may be driven by ‘core’ metabolic processes that are powerfully determined by variations in the mitochondrial melatonergic pathway across different cell types. This provides a different perspective on the nature of local intercellular interactions across diverse ‘autoimmune’-linked conditions, as in the intercellular interactions across different cell types in the pancreatic islets in the pathoetiology of T1DM [17]. This is also relevant to intercellular interactions occurring in the substantia nigra pars compact in Parkinson’s disease [140], and the interactions of astrocytes, microglia, neurons, endothelial cells and pericytes in Alzheimer’s disease [141].

It is upon such ‘core’ mitochondrial, intercellular interactions that genetic, epigenetic, circadian and gut microbiome/permeabilities determine many of their significant effects in the pathoetiology and pathophysiology of a host of diverse medical conditions, many of which have an ‘autoimmune’ aspect. As indicated throughout, the capacity of intercellular processes to regulate the mitochondrial melatonergic pathway is an important aspect of ‘core’ mitochondrial processes in the course of intercellular interactions. This has been most frequently shown to be the case in the tumor microenvironment, where tumor-released kynurenine activates the AhR on NK cells and CD8^+^ T cells to induce a state of ‘exhaustion’ in these tumor-killing cells [112].

Immune cells are ‘invited’ (chemoattracted) into this intercellular interaction when these dynamic intercellular interactions depart too far from homeostasis to challenge the functioning and survival of cell(s) within a given ‘microenvironment’. Intercellular processes within the microenvironment that limit the capacity of a given cell (type) to upregulate the mitochondrial melatonergic pathway drive the suppression of intracrine, autocrine and paracrine melatonin that further upregulates oxidative stress in conjunction with decreased PINK1/parkin and mitophagy, thereby increasing MHC-1, NK cell and CD8^+^ T cell chemoattraction, and ‘autoimmune’-linked cell elimination. Clearly, genetic factors may be acting preferentially in a given cell (type) to bias this outcome of intercellular interactions in a given microenvironment, whilst some cells may be made more liable to ‘autoimmune’-linked elimination by medications, such as the not-uncommon effects of lithium in the induction of autoimmune thyroiditis and hypothyroidism in bipolar disorder patients [142].

An emphasis on the role of intercellular mitochondrial interactions being powerfully determined by variations in the capacity of a given cell (type) to optimize the melatonergic pathway for its own benefit and survival may be seen as parsimonious with the origins of multicellular life on Earth. As noted, the origins of multicellular life emerged around 2 billion years ago when ancient bacteria crept into a single-cellular structure and interacted with a similar bacteria-containing single-cellular organism [104]. This ‘ancient bacteria’, which evolved into mitochondria, seem to have been melatonin-producing [104], with all examples of the three kingdoms of multicellular life over the course of 2 billion years seemingly maintaining the capacity to induce the melatonergic pathway, mostly within mitochondria. As noted, the systemic effects of gut microbiome-derived butyrate and pineal melatonin are via mitochondrial melatonergic pathway upregulation, allowing these systemic processes to optimize mitochondrial function across different cell types and therefore in the maintenance of viable intercellular homeostasis. However, the plasticity of the melatonergic pathway, especially the NAS/melatonin ratio, can also contribute to pathophysiology, including ‘autoimmune’ pathophysiology.

## 6. Integrating Mitochondrial Function across ‘Autoimmunity’

Mitochondrial function is clearly an integral aspect of cells and body systems, with relevance across all medical conditions, including as to how ‘autoimmune’ processes associate with these medical conditions. The human body may be conceptualized as a mass of evolutionary-modified bacteria (in the form of mitochondria) interacting with each other and actual bacteria, including gut, oral and skin bacteria. Higher order body systems, including immune and CNS, are constructed by evolutionary forces from these basic building blocks. The current article highlights the role of mitochondrial function in three aspects of ‘autoimmune’ disorders, namely: (1) intercellular interactions leading to dyshomeostasis in a given microenvironment within different tissues and organs; (2) the attraction and activation of cytolytic cells, including NK cells and CD8^+^ T cells; (3) the regulation of B-cell function, phenotypes and antibody production, as shown in Figure 4 (see Figure 6).

### 6.1. Intercellular Interactions and Dyshomeostasis

Factors acting to dysregulate homeostasis in a given intercellular microenvironment pose an adaptive challenge to the cells within this microenvironment. As with bacteria, mitochondria respond to challenge primarily via oxidant upregulation and release. Mitochondria are the major inducers of oxidants in any given cell, which clearly pose an oxidant challenge within that cell and to the cells in the immediate environment. This oxidant challenge is typically met by the upregulation of antioxidants and antioxidant enzymes, of which the mitochondrial melatonergic pathway is a significant aspect. Factors acting to suppress mitochondrial ATP from OXPHOS and the TCA cycle via the suppression of sirtuin-3/Bmal1 disinhibition of the PDC conversion of pyruvate to acetyl-CoA limit the capacity of cells to induce the melatonergic pathway, given that acetyl-CoA is a necessary precursor for the conversion of serotonin to NAS. The mitochondrial melatonergic pathway can be inhibited by a number of means, including the suppression of tryptophan and serotonin availability and uptake, as well as the suppression of tryptophan hydroxylase (TPH) and the specific 14-3-3 isoforms that are necessary to stabilize TPH and AANAT [17]. A wide array of genetic, epigenetic and intercellular fluxes may therefore bias the capacity of a given cell to optimize its mitochondrial function and resistance to oxidant challenge via impacts on the cell’s ability to upregulate the tryptophan-melatonin pathway.

Suppression of the tryptophan-melatonin pathway enhances mitochondrial oxidants, which induce ROS-dependent miRNAs, leading to changes in patterned gene transcription and fluxes within the local microenvironment (see Figure 6). This is an important mechanism in the local intercellular dysregulation occurring across classical ‘autoimmune’ and associated disorders. Notably, melatonergic pathway induction does not necessarily drive melatonin production. The upregulation of the NAS/melatonin ratio by the AhR, P2Y1r and mGluR5 [114,115] provides NAS to activate TrkB-full length (TrkB-FL) and TrkB-T1. Although TrkB-T1 (and TrkB-FL) seem to have some beneficial effects in pancreatic β-cells in T1DM, the activation of TrkB-T1 is typically associated with apoptotic processes, primarily from preventing the trophic and metabolic effects of BDNF. It would be parsimonious with such pro-apoptotic effects of TrkB-T1 should its induction be intimately linked to suboptimal mitochondrial function, perhaps via the upregulation of miRNAs that enhance TrkB-T1 levels, such as miR-34a and miR-4813, or the suppression of miR-185 [143]. However, this requires investigation across diverse ‘autoimmune’ disorders. The induction of TrkB-T1, as at the neuromuscular junction in ALS, may be an important suppressor of autocrine and paracrine trophic support that is intimately linked to mitochondrial dysfunction in the course of ‘autoimmune’ etiology in ALS and across different autoimmune disorders [15].

As well as its role in the suppression of melatonin availability and the upregulation of the NAS/melatonin ratio, leading to the NAS activation of TrkB-T1, the AhR has several roles in the pathophysiology of ‘autoimmune’ disorders, including by increasing programmed cell death (PD)-1 expression in different cell types [144,145], including CD8^+^ T cells [146]. This has well-established relevance in the tumor microenvironment [112], as well as to the elevated PD-1 levels in Alzheimer’s disease, which correlate with suppressed cognition [147]. The raised levels of AhR and AhR ligands are evident over the course of aging, contributing to immune dysregulation in association with impaired mitochondrial function [148]. It will be important to determine whether the suppression of the mitochondrial melatonergic pathway, including by the AhR suppression of melatonin, is a significant determinant of ROS-driven miRNAs that regulate PD-1 and its ligand, PD-L1, coupled to AhR-driven NAS and raised TrkB-T1 levels.

The interface of the mitochondrial melatonergic pathway with miRNAs is a cutting-edge area of research relevant to ‘autoimmune’-linked processes. For example, miR-138 suppresses PD-1, and another ‘immune-checkpoint’ inhibitor, cytotoxic T-lymphocyte-associated molecule 4 (CTLA-4) [149]. The role of the mitochondrial melatonergic pathway in miR-138 regulation is indirectly indicated by data showing mitophagy upregulation to enhance miR-138 via promotor demethylation, with miR-138 suppressing the NLRP3 inflammasome, as well as PD-1 and CTLA-4 [150]. The capacity of melatonin, both pineal and locally, to upregulate mitophagy as well as to suppress oxidative stress and the NLRP3 inflammasome in Alzheimer’s disease and Alzheimer’s disease models [151] would indicate that melatonin upregulates miR-138, thereby contributing to NLRP3, CTLA-4 and PD-1 suppression, with relevance to aging-associated changes in immune responsivity (see Figure 7). As with other cell types, the regulation of the mitochondrial melatonergic pathway in immune and glia cells will be important to determine over the course of aging, especially whether the 10-fold decrease in pineal melatonin production and release at 80 years, compared to 18 years, is replicated in other cell types. Whether aging-associated immune changes limit the likelihood of developing a classical ‘autoimmune’ disorder during aging, as is the case in T1DM [17], but contribute to the maintenance of dyshomeostasis in local microenvironment interactions, requires investigation across different medical conditions. This could suggest that changes over aging limit the capacity to develop a full-blown classical ‘autoimmune’ disorder, but with the dysregulated homeostasis being the basis of aging-linked conditions, such as neurodegenerative disorders. Overall, alterations in the mitochondrial melatonergic pathway and mitophagy during aging will impact intercellular homeostatic processes in the microenvironment of specific medical conditions, with aging-associated increases in AhR and ligands altering the interactions of cytolytic cells within these microenvironments.

The growing appreciation of the gut microbiome and pineal melatonin in the biological underpinnings of a host of diverse medical conditions seems importantly determined by the effects of butyrate and pineal melatonin on mitochondrial function, including via the upregulation of the mitochondrial melatonergic pathway across many body cells [100]. When optimal, both butyrate and pineal melatonin optimize mitochondrial function and the induction of the mitochondrial melatonergic pathway, thereby suppressing oxidants and inflammation during the maintenance of homeostatic processes within a given microenvironment [118,122]. Significant pathogen challenges, via pro-inflammatory cytokine upregulation, suppress pineal melatonin production and increase gut permeability, thereby enhancing gut dysbiosis, with associated butyrate suppression. Factors causing a pre-existent change in the gut microbiome, such as the glyphosate-based herbicide suppression of the *Akkermansia muciniphila* and the shikimate pathway, lead to gut permeability and decreased butyrate. This suppresses the gut microbiome’s capacity to optimize mitochondrial function and makes established homeostasis across the body more susceptible to dysregulation. Susceptibility to a host of diverse medical conditions with ‘autoimmune’ aspects is increased because of gut dysbiosis/permeability.

*Akkermansia muciniphila* is the major determinant of the shikimate pathway in the human gut [102], thereby determining tryptophan production by the shikimate pathway, with consequences for tryptophan conversion to tryptamine to activate the AhR and maintain the gut barrier. Suppressed *Akkermansia muciniphila* is associated with both classical and newly conceptualized ‘autoimmune’ disorders, including T1DM [152], and multiple sclerosis [153] as well as amyotrophic lateral sclerosis [15], Alzheimer’s disease [154] and Parkinson’s disease [155]. Although most tryptophan is derived from dietary sources, the regulation of tryptophan production by the shikimate pathway and *Akkermansia muciniphila* has pathophysiological relevance, especially under conditions where pro-inflammatory cytokine induction of IDO drives the conversion of tryptophan to kynurenine and therefore away from serotonin, NAS and melatonin production. Such data highlight the relevance of the gut microbiome to the regulation of the tryptophan-melatonin pathway as well as to coordinated changes in mitochondrial function and microenvironment homeostasis (see Figure 6).

Gut *Lactobacillus johnsonii* is also commonly decreased in ‘autoimmune’ disorders, where its administration affords some benefits [156]. *Lactobacillus johnsonii* increases gut microbiome short-chain fatty acids, including butyrate, in the challenged gut [157] as well as suppressing levels of the fungi *Candida albicans* [158]. *Candida albicans* has an increased presence in the gut and oral microbiomes as well as systemically in an array of ‘autoimmune’-linked conditions, including multiple sclerosis, where *Candida albicans*-induced IL-23 activates mucosal-associated invariant T (MAIT) cells, with MAIT cells enhancing pro-inflammatory processes, including in the CNS [159]. *Candida albicans* is also a significant driver of T1DM [160], whereas *Lactobacillus johnsonii* has treatment efficacy [161]. The therapeutic efficacy of *Lactobacillus johnsonii* seems predominantly via the suppression of *Candida albicans*, as well as the suppression of gut dysbiosis coupled to increased butyrate and other short-chain fatty acids that can act to optimize mitochondrial function throughout the body. Wider gut microbiome dysregulation by bacteriophages and enteroviruses also contributes to ‘autoimmune’ disorders [17] (see Figure 6).

Overall, the intercellular interactions of mitochondria in the local microenvironment in a given organ/tissue are crucial to the pathoetiology of ‘autoimmunity’. Systemic processes, such as gut dysbiosis/permeability and suppressed pineal melatonin, can contribute to local intercellular dyshomeostasis via impacts on the mitochondrial melatonergic pathway. It is the alterations in mitochondrial function that dysregulate intercellular interactions leading to an increase in the oxidative stress-induced suppression of PINK1 and mitophagy, thereby further contributing to mitochondrial energy and oxidant dysregulation, leading to raised MHC-1 levels and the chemoattraction of CD 8^+^ T cells.

### 6.2. Cytolytic Cells and Mitochondrial Dysfunction

As indicated above, alterations in the intercellular interactions of mitochondria across cell types can trigger a dyshomeostasis that leaves a particular cell (type) compromised in its capacity to regulate the oxidant/antioxidant balance, likely involving the suppression of the tryptophan-melatonin pathway. The resultant suppression of melatonin, PINK1/parkin and mitophagy leads to the chemoattraction of NK cells and CD8^+^ T cells in response to MHC-1 induction. The enhanced pro-inflammatory cytokine induction acts to increase gut permeability and associated gut dysbiosis as well as suppress pineal melatonin, thereby allowing cytolytic cells to resolve the dyshomeostasis [119].

In classical ‘autoimmune’ disorders, such as T1DM, the initial activation of pancreatic islet-specific CD8^+^ T cells occurs in the pancreatic lymph nodes [44]. This is a standard description of an autoimmune disorder. However, these authors also show, in a T1DM preclinical model, that the dysregulation in the gut microbiome is important in driving a ‘bystander’ enhancement of effector function in CD8^+^ T cells [44], indicating that variations in the gut microbiome are an integral aspect of the strength of ‘autoimmune’ responses, with relevance to initial symptom severity [44]. This is supported by further data in this study showing that the oral administration of the short-chain fatty acid, butyrate, suppresses the additional effector function in the gut’s Peyer’s patches, thereby attenuating the non-specific bystander potentiation of cytotoxicity in autoreactive CD8^+^ T cells [44]. A number of studies have shown the gut microbiome’s short-chain fatty acids, including propionate and acetate, but especially butyrate, to modulate the effector and memory functions of CD8^+^ T cells [162,163]. The effects of butyrate on CD8^+^ T cells seem mediated via impacts on metabolism and mitochondrial function, being another example of how variations in gut bacteria communicate with, and modulate, evolutionary-modified bacteria in the form of mitochondria [100]. The gut may also induce activation of the CD8^+^ T cells via antigen priming [87], and exosome release [164], indicating the importance of variations in the gut microbiome and its interface with the host immune response.

As noted, aging-associated increases in the AhR and its ligands may contribute to the heightened PD-1, PD-L1 and CTLA-4 associated with mitophagy and suppressed tryptophan-melatonin pathway availability. As well as the short-chain fatty acids, the gut microbiome is a ready source of numerous tryptophan-derived AhR ligands, such as indole-3-propionate. This requires investigation regarding the differential effects of gut microbiome/permeability in cytolytic cell regulation, including AhR regulation of the mitochondrial melatonergic pathway and NAS/melatonin ratio in CD8^+^ T cells, and the consequences that this has for CD8^+^ T cell effector and memory functions. The aging-associated suppression of the naïve CD8^+^ T cell function seems mediated via suppressed metabolism [165], and the interface of mitochondrial OXPHOS with the glycolytic upregulation that is necessary for CD8^+^ T cell activation [62].

The AhR and its ligands have been proposed to underpin many aging-associated changes, including in immune responsivity [166]. The capacity of raised AhR levels and ligands, as well as mitophagy, to upregulate immune checkpoint inhibitors in cytolytic cells will be important to determine, both at the site of tissue dyshomeostasis and in the gut. As AhR activation increases the NAS/melatonin ratio and ‘backward’ conversion of melatonin to NAS via O-demethylation, the concurrent upregulation of TrkB-T1 by ROS-driven miRNAs in ‘autoimmune’-challenged cells may make heightened AhR levels and ligands important determinants of aging-associated changes in ‘autoimmune’ processes. Whether such aging-associated increases limit the capacity of CD8^+^ T cells and NK cells to drive a heightened, classical ‘autoimmune’ disorder, out of which emerges neurodegenerative conditions, such as Parkinson’s disease and Alzheimer’s disease, will be interesting to determine.

### 6.3. B-Cells and Metabolism

Unlike most immune cells, B-cells do not upregulate glycolysis in the course of activation and are dependent upon increasing ATP production by OXPHOS and the TCA cycle to produce cytokines, antibodies and present antigens to T cells [117,118]. Mitochondrial function and mitophagy are crucial to survival in circulating memory B-cells, during which many systemic factors can impact B-cell function with relevance to suboptimal memory B-cell function during aging [80,82] (see Figure 4).

AhR activation significantly regulates B-cell function, with AhR levels in B-cells positively correlating with disease activity in SLE, especially SLE with renal damage [167]. In the experimental autoimmune encephalitis (EAE) model of multiple sclerosis, 2,3,7,8-tetrachlorodibenzo-p-dioxin (TCDD) activation of the AhR affords protection by suppressing IgG1 and IgG3 in splenocytes and B-cells, thereby suppressing the recruitment of cytolytic cells [168]. Other data indicate that AhR activation by TCDD can induce regulatory functions in B-cells, without shifting these cells to a classical Breg phenotype [169]. This is evident in lung cancer where myeloid-derived suppressor cells’ (MDSCs) induction of IDO leads to kynurenine that activates the AhR to induce Breg and immune suppression [170]. In long-lived plasma cells, the kynurenine activation of the AhR is essential to their longevity and function [171]. Overall, such data highlight the relevance, and complexity, of AhR effects in humoral immunity. This complexity can arise from the differential effects of different AhR ligands as well as the capacity of the AhR in each cell to regulate the melatonergic pathway and NAS/melatonin ratio, especially given the role of TrkB in B-cell function [88]. However, given the importance of mitochondrial function to B-cells, it is notable that the AhR can also be expressed on the mitochondrial membrane, as can TrkB, with mitochondrial AhR activation regulating Ca^2+^ influx via the voltage-dependent anion channel (VDAC)1 and regulating mitochondrial transcription [172,173]. The VDAC is expressed on the mitochondrial as well as on the plasma membrane of B-cells [174].

Clearly, the role of the mitochondrial AhR in B-cells and other cells during mitochondrial dysregulation in ‘autoimmunity’ requires investigation, including whether the mitochondrial AhR can act to regulate the mitochondrial NAS/melatonin ratio and the consequences that this has for cellular function and intercellular communication. The capacity of the array of endogenous and exogenous AhR ligands to activate the mitochondrial AhR may be a contributory factor to the diverse data arising from investigations of AhR activation, including in the course of autoimmune disorders and aging. This may be especially important in B-cells given their dependence on the transcription factors NF-kB and YY1 that are intimately linked to the upregulation of the mitochondrial melatonergic pathway, with YY1 crucial to CSR in activated B-cells [93].

Overall, there is still a predisposition to favor subtyping immune cells, including B-cells, on the basis of plasma membrane expressions, at the expense of looking at the complexity of mitochondrial processes that are generally regarded as crucial to variations and gradations in cellular function. The direct impacts of the AhR and TrkB on mitochondrial function and how these two receptors interface with the mitochondrial melatonergic pathway should better clarify, and further highlight, the roles of mitochondrial function and their intercellular interactions in the course of ‘autoimmunity’. The above also has implications for how ‘autoimmune’ disorders’ interface with mood and wider psychiatric conditions.

## 7. Autoimmunity and Psychiatric Disorders

The pathophysiological underpinnings of ‘autoimmune’/‘immune-mediated’ disorders overlap with the pathophysiology of most psychiatric conditions, such as mood/anxiety disorders, bipolar disorder and schizophrenia, including increased O&NS [175], suboptimal mitochondrial function [176], circadian dysregulation [177], gut dysbiosis [178] and tryptophan-melatonin pathway dysregulation [4,179] as well as alterations in NK cells and CD8^+^ T cells [180]. Classical ‘autoimmune’ disorders, such as multiple sclerosis [181], T1DM [182] and rheumatoid arthritis [183] show elevations in concurrent psychiatric presentations, especially major depressive disorder (MDD).

MDD modulates the initiation and/or episodic symptom exacerbations of most ‘autoimmune’/‘immune-mediated’ disorders, including multiple sclerosis [184], rheumatoid arthritis [183], Alzheimer’s disease [185], Parkinson’s disease [186] and SLE [187], indicating the role that MDD pathophysiology has in their course. MDD pathophysiology includes gut dysbiosis/permeability, pro-inflammatory cytokines, IDO induction and the conversion of tryptophan to kynurenine, leading to enhanced AhR activation, thereby dysregulating the tryptophan-melatonin pathway [188], and allowing the MDD pathophysiology to be intimately linked to the pathophysiological processes underpinning ‘autoimmune’ disorders. The long-standing association of decreased serotonin in MDD is closely intertwined with lower serotonin availability as a precursor for the melatonergic pathway, and therefore intimately linked to alterations in mitochondrial function and cellular plasticity, especially under challenge [188]. Notably, MDD pathophysiology seems highly heterogenous [189], with astrocytes being an important hub in coordinating changes in neuronal activity and interarea patterned neuronal activity in the brain [4,179]. However, MDD samples in any given study show significant heterogeneity in physiological indices, the investigation of which should better define the classification and treatment not only of MDD [98,190], but of the associations of MDD with autoimmune disorders.

In a nationwide survey, experiencing an MDD episode increased the subsequent risk of many ‘autoimmune’ disorders, including psoriasis, lichen planus, alopecia areata, morphea, autoimmune bullous diseases, hidradenitis suppurativa, vitiligo, SLE, systemic sclerosis, Sjogren’s syndrome and dermatomyositis [191]. The role of dysregulation in the mitochondrial melatonergic pathway of these predominantly autoimmune skin diseases requires investigation, given that the serotonergic and melatonergic pathways are fully expressed in human skin cells [192]. Skin disorders are often evident in T1DM, usually in association with heightened levels of advanced glycation end-products (AGEs) [193]. As AGEs act via the AGE receptor, RAGE, to increase NF-kB and pro-inflammatory signaling in the skin as in other body sites [194], the attenuated capacity to upregulate the mitochondrial melatonergic pathway in the T1DM skin will be important to determine. Skin disorders are common in other ‘autoimmune’/‘immune-mediated’ disorders, including multiple sclerosis [195] and Parkinson’s disease [196].

Given the association of MDD with autoimmune disorders, the suboptimal mitochondrial function and the dysregulated tryptophan-melatonin pathway, it has been proposed that some MDD presentations may be conceptualized as ‘autoimmune’ disorders, as with MDD presentations arising from subclinical hypothyroidism driven by thyroid autoimmunity [197]. These authors showed autoimmune thyroiditis to be significantly positively correlated with MDD and anxiety scores as well as metabolic syndrome and suicide attempts [197]. Interestingly, the thyroid-stimulating hormone induces the melatonergic pathway enzymes in thyroid C-cells, which release melatonin to regulate the thyroid gland function, including by the direct regulation of thyroglobulin gene expression in follicular cells [198]. This indicates that thyroid autoimmunity, including its association with MDD, may be linked to alterations in the regulation of the mitochondrial melatonergic pathway in the thyroid gland. As noted, autoimmune thyroiditis and hypothyroidism are not uncommon in bipolar disorder patients treated with lithium [142], overlapping thyroid autoimmune processes with wider mood presentations and their treatment.

Importantly, MDD correlates with alterations in the two systemic hubs most associated with mitochondrial regulation, namely the gut microbiome/permeability and circadian rhythm [4,179]. Suppressed levels of gut microbiome butyrate, raised LPS levels and suppressed pineal melatonin production are common in MDD [4,179], and, as highlighted above, are integral pathophysiological aspects of ‘autoimmune’/‘immune-mediated’ conditions. MDD-mediated changes in the gut will enhance the ‘bystander’ activation of CD8+ T cells [17,89], allowing MDD associated with gut dysbiosis and decreased butyrate production to modulate core aspects of autoimmune pathophysiology. Likewise, the suppression of pineal melatonin in MDD and other psychiatric conditions will attenuate the capacity of night-time melatonin to dampen residual inflammatory activity and optimize mitochondrial function, thereby helping to maintain pre-established intercellular homeostatic interactions across body microenvironments. It will be important to determine as to whether it is the gut dysbiosis/permeability in MDD pathophysiological heterogeneity that underpins the association of a particular MDD pathophysiological subtype with ‘autoimmune’/‘immune-mediated’ conditions.

Notably, as well as MDD, other psychiatric conditions, including bipolar disorder and schizophrenia, also show evidence of increased gut dysbiosis/permeability and circadian dysregulation [177,199], indicating pathophysiological overlaps with ‘autoimmune’/‘immune-mediated’ conditions. Interestingly, bipolar disorder, schizophrenia and MDD show changes in pathophysiological underpinnings over time, determined partly by the number of recurrent episodes and treatment, termed neuroprogression [200]. Neuroprogressive changes in MDD and bipolar disorder have recently been proposed to underpin the increased risk of dementia in people diagnosed with MDD and bipolar disorder, and of Parkinson’s disease risk in bipolar disorder [201]. Mood disorders and more severe psychiatric disorders are commonly associated with insomnia and sleep dysregulation [202]. Recent work shows insomnia and sleep dysregulation to be intimately linked to immune dysregulation, especially the upregulation of low-level pro-inflammatory cytokines, and changes in the levels and activity of NK cells [203]. This is parsimonious with the loss of melatonin’s immune-dampening effects at night, and its effects on mitochondrial function in immune cells. Overall, the gut, circadian and neuroprogressive changes in psychiatric disorders are intimately linked to the pathophysiology of ‘autoimmune’/‘immune-mediated’ disorders, with the data highlighted throughout this article outlining how this may occur (see Figure 8).

The pathophysiological underpinnings of ‘autoimmune’/‘immune-mediated’ conditions and their overlaps with neuropsychiatric conditions pose significant challenges for the current conceptualizations, classification and treatment of these conditions. The data highlighted above have significant implications for future research and treatment, as indicated below.

## 8. Future Research Directions

Given the importance of mitochondrial function and oxidative stress-induced mitochondrial DNA mutations in B-lymphocytes [92], it clearly requires investigation as to the presence and expression of the mitochondrial melatonergic pathway in B-cells, CD8^+^ T cells and NK cells.

Does pineal melatonin regulate the function and survival of circulating memory B-cells, suggesting that the 10-fold decrease in pineal melatonin from the ages of 18 years to 80 years will contribute to aging-linked variations in antibody response and wider ‘autoimmune’-associated processes? Is the 10-fold decrease in pineal melatonin over aging replicated in other cell types? If so, this could significantly change the understanding of aging and how it interfaces with almost all medical conditions.

Does the suppression of night-time, pineal melatonin modulate the likelihood of a classical autoimmune response across age groups? Would this have relevance in cancers, such as breast cancer which is linked to mitochondria autoimmunity in its inception and progression [204]?

It will be important for future research to clarify how gut-derived LPS interacts with mitochondrial membrane factors in the regulation of NF-kB signaling under inflammatory conditions. Given the presence of a 14-3-3-like motif on the matrix tail of mitochondria-membrane located, LETM1, which may bind 14-3-3 and/or AANAT in the proximity of mitochondrial ribosomes, the LUBAC upregulation of NF-kB signaling may be a crucial factor in the induction of the deactivation process via the PINK/LETM1/melatonergic pathway induction in reactive cells. As oxidative stress suppresses PINK1 levels, thereby decreasing mitophagy [123], the capacity to upregulate the melatonergic pathway may be an integral aspect of how cells manage inflammatory processes. This will be important to clarify in future research across different cell types.

In consideration of the growing appreciation of the importance of enteric glia in not only regulating the enteric nervous system, but also being an important interface between the gut microbiome and the mucosal immune system and vagal nerve, is there anything to indicate that a suppressed mitochondrial melatonergic pathway function is evident in enteric glia cells, leading to attenuated PINK1 expression under the conditions of raised O&NS? This would parallel changes in the interface of astrocytes and dopaminergic neurons in the substantia nigra pars compacta in Parkinson’s disease, which may be parsimonious with enteric glia as gut astrocytes.

There is an increasing interest in the role of the alpha 7 nicotinic acetylcholine receptor (α7nAChR) in classical ‘autoimmune’ pathophysiology and treatment, including rheumatoid arthritis [205]. This overlaps with the vagal nerve release of acetylcholine and the use of vagal nerve stimulation in the management of some classical autoimmune disorders, such as SLE [206]. Of note, melatonin upregulates the α7nAChR [207], with melatonin’s maintenance of the gut barrier mediated via the α7nAChR [208]. The interface of the vagal nerve, α7nAChR and gut barrier with pineal and mitochondrial melatonin will be important to determine, with possible treatment implications.

What underpins the vagal nerve induction of a TrkB ligand and associated beneficial effects in preclinical models of anxiety and depression [209]? Does vagal nerve stimulation increase vagal NAS production via impacts on the vagal mitochondrial melatonergic pathway and/or on the cells onto which vagal ACh is released? Is this linked to activation of the a7nAChR? Should the vagal nerve release NAS, would this suggest that processes acting to upregulate TrkB-T1 in a given cell would then allow vagal nerve stimulation to contribute to that cell’s elimination? Would a cell challenged by such putative vagal NAS at TrkB-T1 act to suppress vagal activity? Would the autonomic dysregulation common across many autoimmune disorders be linked to this?

NAS, similar to BDNF, activates the TrkB-FL, leading to trophic, proliferative and generally protective effects, whilst TrkB-T1 is generally associated with decreased trophic support to cells, thereby contributing to apoptosis [210,211]. Many ‘autoimmune’ disorders show upregulated TrkB-T1 and/or a dependence of cells or immune processes on BDNF. It will be important to determine whether the suppression of the mitochondrial melatonergic pathway and associated increase in ROS-induced miRNAs include miRNAs that can regulate TrkB-T1, such as miR-34a, miR-4813 and miR-185, therebyco-ordinating the upregulation of the NAS/melatonin ratio with TrkB-T1 [143].

It will also be important to determine whether TrkB-T1 regulation is driven by cells in the local microenvironment, akin to the original conception of TrkB-T1 as a mediator of ‘competitive elimination’ in the course of excessive neuron levels in early development.

Given the clear importance of mitochondrial function across all relevant cells in ‘autoimmune’ processes, it will be important to investigate the presence and effects of the AhR, TrkB-FL, TrkB-T1 and α7nAChR on the mitochondrial membrane [172,173,212], and how these interface with the mitochondrial melatonergic pathway. This will also be important to determine during intercellular interactions and processes.

Is the association of TrkB-T1 with raised Ca^2+^ influx and insulin secretion in pancreatic β-cells [211] mediated via TrkB-T1 on the mitochondrial membrane? Does the miRNA/ROS induction of TrkB-T1 differentially induce TrkB-T1 expression in mitochondria?

It will be important to determine whether the suppression of the mitochondrial melatonergic pathway, including by the AhR suppression of melatonin, is a significant determinant of ROS-driven miRNAs that upregulate PD-1 and its ligand PD-L1. Is this coupled to AhR-driven NAS and raised TrkB-T1 levels?

Is the AhR expressed on the mitochondrial membranes of CD8^+^ T cells, NK cells and other cells relevant across different medical conditions? Is there a variable capacity of the wide array of endogenous and exogenous AhR ligands to activate the mitochondrial AhR, with consequent differential impacts on mitochondrial function and transcription?

There is a growing interest in the role of exosomes, including exosome-containing MHC-1, in the priming of CD8^+^ T cells, predominantly investigated in tumor cell-derived exosomes [213]. Several factors regulate MHC-1 levels in exosomes, including HDAC and NF-kB signaling [214]. The relevance of exosomal MHC-1 in classical and wider ‘autoimmune’ disorders will be important to determine, as will other exosomal contents that can suppress cytolytic cells, including TGF-β-upregulating exosomal PD-L1 [215].

The adoptive transfer of mouse B-cell-derived exosomes drives CD8^+^ T cell responses, whilst metabolically overactive B-cells enhance autoimmune processes, including from the expression of exosomal MHC [216]. Intestinal epithelial cells may also release exosomes containing MHC [164], suggesting another mechanism whereby variations in the gut microbiome interactions with intestinal epithelial cells may prime classical ‘autoimmunity’. The relevance of these exosomal releases to the mitochondrial melatonergic pathway and wider mitochondrial function requires investigation.

Astrocyte MHC-1 upregulation, likely via exosomal release, contributes to heightened microglia inflammatory activity and neuronal atrophy [217]. Does the suppression of the melatonergic pathway in astrocytes in dementia [13] contribute to raised exosomal MHC-1 release from astrocytes? Given that enteric glial cells are ‘gut astrocytes’, is MHC-1 released by enteric glial cells to prime ‘autoimmune’-linked processes?

Adverse childhood events/abuse are long associated with MDD pathoetiology and distinct pathophysiology [218]. Recent work indicates that specific alterations in the gut microbiome in MDD lead to distinct pathophysiology-determined phenotypes [219]. This will be important to further investigate, especially given the association of childhood abusive events/abuse with ‘autoimmune’/‘immune-mediated’ disorders [220].

Does vagal nerve acetylcholine activate the mitochondria located α7nAChR in enteric glia cells to regulate exosomal MHC-1 content?

Importantly, mitochondria have multi-faceted roles, including in providing the building blocks for a number of biosynthetic pathways, including fatty acids, cholesterol, glucose, amino acids and heme [221]. Although cholesterol levels are not high in the inner mitochondrial membrane, mitochondrial cholesterol is important in different organs for the biosynthesis of neurosteroids in neurons and hepatic bile acids [222]. Interestingly, the mitochondrial ATPase, ATPase Family AAA Domain Containing 3A (ATAD3A), is not only crucial to cholesterol transport and mitochondrial structure, but also cell survival, with ATAD3A knockout suppressing PINK1 and mitophagy levels in association with mitochondrial damage linked to raised free cholesterol levels [223]. As melatonin seems to increase mitochondrial membrane fluidity/lipid rafts [120], it requires an investigation as to how variations in mitochondrial melatonergic pathway regulation modulate the building blocks for the biosynthesis of general and organ/tissue/cell-specific products, such as neurosteroids in neurons. Notably, butyrate as well as melatonin are significant regulators of cholesterol and fatty acid metabolism [224,225]. Any differential effects of NAS at TrkB-FL and TrkB-T1 on the plasma membrane and mitochondrial membrane, versus melatonin, on ATAD3A and mitochondrial cholesterol will be important to determine.

## 9. Treatment Implications

Several treatment implications will be better defined following the above proposed research.

Clearly, targeting the optimization of the mitochondrial melatonergic pathway from the refinement of, e.g., mesenchymal stem cell vesicular/exosomal contents will potentially have wide-acting implications in the prevention, treatment and re-establishment of homeostatic interactions across classical and wider ‘autoimmune’ disorders.

Is there wider utility from the utilization of α7nAChR agonists, as indicated in rheumatoid arthritis [205]?

Will targeting particular or specific collections of gut bacteria prove the utility in the prevention as well as treatment of an array of ‘autoimmune’-linked disorders, including via the upregulation of *Akkermansia muciniphila* and *Lactobacillus johnsonii*, coupled to the suppression of *Candida albicans*?

## 10. Conclusions

There is a growing appreciation as to the importance of mitochondrial function across all medical conditions, including how ‘autoimmune’-associated processes couple with diverse medical presentations. This article has highlighted how the tryptophan-melatonin pathway in driving the mitochondrial melatonergic pathway is an important aspect of dysregulated intercellular homeostasis underpinning most medical conditions, and how such intercellular mitochondrial interactions can drive cytolytic cell and B-cell processes classically seen to underpin ‘autoimmunity.’ This has relevance to a wide range of poorly conceptualized and treated conditions, such as T1DM, ALS, SLE, Parkinson’s disease and Alzheimer’s disease. Systemic processes known to regulate a wide array of diverse medical presentations, including the gut microbiome/permeability and circadian rhythm, can be seen as having their impacts on mitochondrial function and the mitochondrial melatonergic pathway. The future research directions indicated should provide novel treatments that are targeted to optimizing ‘core’ physiological processes, of which the tryptophan-melatonin pathway and mitochondrial melatonergic pathway are crucial, overlapping elements.

## Figures and Tables

**Figure 1 cells-12-01237-f001:**
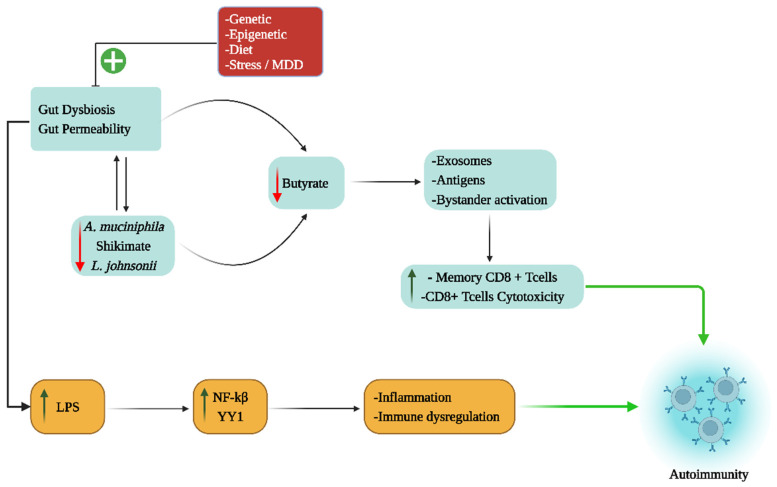
The figure shows how systemic processes such as the gut microbiome/permeability can influence the immune response. Genetic, epigenetic, dietary and stress/MDD can increase gut permeability-derived LPS, leading to an increase in the pro-inflammatory transcription factors, NF-kB and YY1, to contribute to inflammation and patterned immune response dysregulation that influence processes driving autoimmune responses. Gut dysbiosis can be induced by genetic, epigenetic, dietary and stress/MDD processes, including via a decrease in the operation of the Shikimate pathway and associated suppression of the gut bacteria, *Akkermansia muciniphila* and *Lactobacillus johnsonii*, leading to a decrease in the HDACi and mitochondrial-optimizing effects of the short-chain fatty acid, butyrate. Butyrate suppresses the exosomes, gut antigen priming and bystander activation of CD8^+^ T cells, thereby suppressing the induction of ‘autoimmune’ processes. Abbreviations: HDACi: histone deacetylase inhibitor; LPS: lipopolysaccharide; MDD: major depressive disorder; NF-kB: nuclear factor kappa-light-chain-enhancer of activated B-cells; YY1: yin yang 1.

**Figure 2 cells-12-01237-f002:**
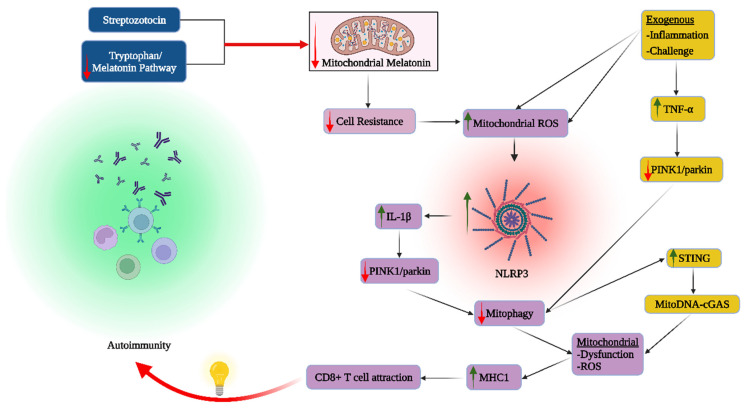
The figure shows how a decrease in melatonin and the melatonergic pathway by streptozotocin or a suppressed tryptophan-melatonin pathway increases cell susceptibility to challenge, including intercellular within a given microenvironment, resulting in increased mitochondrial ROS, which induces the NLRP3 inflammasome to increase IL-1β, which, similar to TNFα, suppresses PINK1/parkin to decrease mitophagy, thereby leading to mitochondrial energy, ROS and ROS-driven microRNA dysregulation, in turn driving raised MHC-1 levels and CD8+ T cell attraction. TNFα inhibition of PINK1/parkin can also increase STING and mitochondrial DNA binding with cGAS. Both processes (purple and gold shading) contribute to ‘autoimmunity’. Abbreviations: cGAS: cyclic guanosine monophosphate-AMP synthase; IL-1: interleukin-1; MHC: major histocompatibility complex; NLRP3: NLR family pyrin domain containing 3; PINK1: PTEN-induced kinase 1; STING: (cGAS)-stimulator of interferon genes; ROS: reactive oxygen species; TNF: tumor necrosis factor.

**Figure 3 cells-12-01237-f003:**
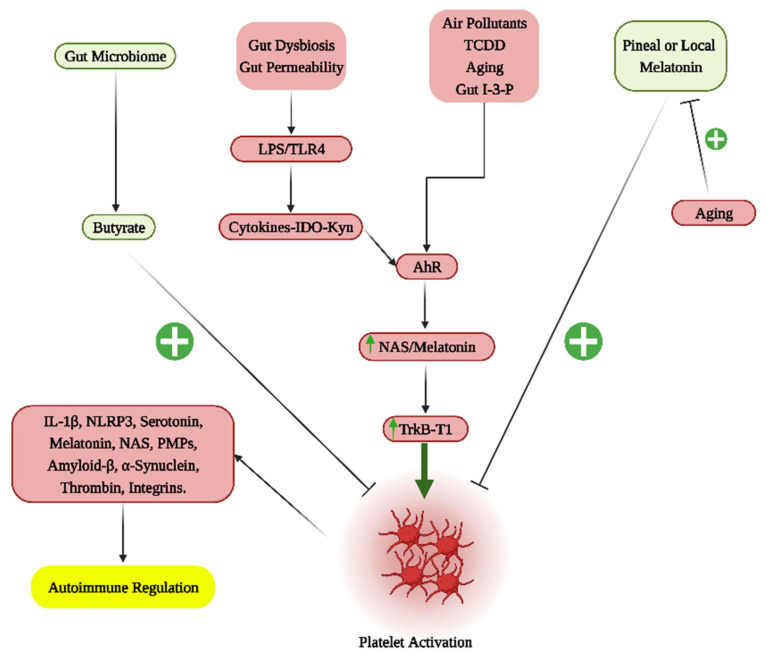
The figure shows how the variety of fluxes following platelet activation can regulate metabolic and immune processes underpinning ‘autoimmune’ disorders, as well as how platelet activation may be regulated, and integrated, with systemic processes, including the gut microbiome/permeability and circadian, pineal melatonin (green shade inhibitory and red shade stimulatory regarding platelet activation). Butyrate production by a healthy gut microbiome inhibits platelet activation, as does pineal or local melatonin. The dramatic decrease in pineal melatonin over aging will suppress the capacity of the circadian rhythm to dampen platelet activation. Gut dysbiosis/permeability increases circulating LPS that activates TLR4 to induce pro-inflammatory cytokines, which increase IDO. IDO conversion of tryptophan to kynurenine activates the AhR, which induces CYP1A2 and CYP1B1 to backward convert platelet melatonin to NAS. NAS, intracrine and/or autocrine can activate platelets via truncated receptor, TrkB-T1. Other AhR ligands, including air pollutants, TCDD and gut-derived I-3-P and its derivates also activate the AhR. Aging increases the AhR, which is expressed in the cytosol and on the mitochondrial membrane. Abbreviations: AhR: aryl hydrocarbon receptor; I-3-P: indole-3-propionate; IDO: indoleamine,2,3-dioxygenease; kyn: kynurenine; LPS: lipopolysaccharide; NAS: N-acetylserotonin; PMPs: platelet microparticles; TCDD: 2,3,7,8-tetrachlorodibenzo-p-dioxin; TLR: toll-like receptor; trkB-T1: tyrosine receptor kinase B-truncated 1.

**Figure 4 cells-12-01237-f004:**
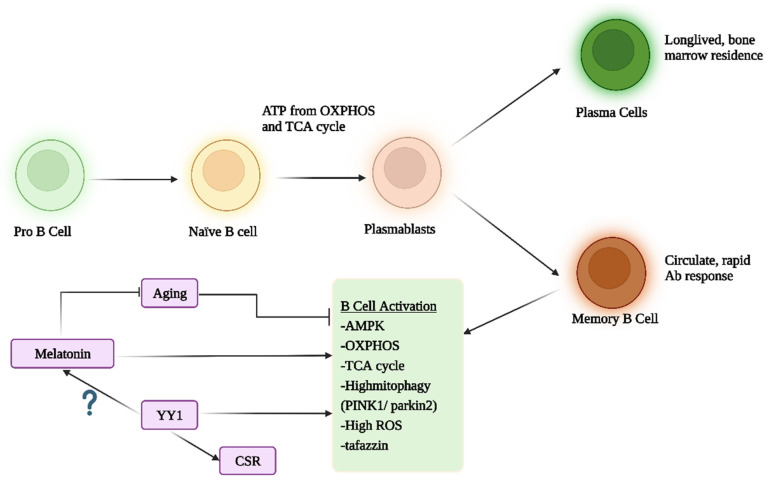
The figure shows B-cell development and the important role of mitochondrial metabolism in the activation, and antibody production, of naïve B-cells and memory B-cells. B-cell activation requires the upregulation of AMPK, OXPHOS, TCA cycle, mitophagy and tafazzin, in association with high ROS production. Aging suppresses B-cell activation and antibody production via ROS upregulation and NF-kB activation, which melatonin suppresses. YY1 contributes to B-cell activation and is crucial to the CSR antibody plasticity response. YY1 induces the melatonergic pathway in other cells, suggesting that melatonin production and availability may be important to B-cell activation, antibody production and CSR plasticity. Abbreviations: Ab: antibody; AMPK: adenosine monophosphate-activated protein kinase; CSR: class switch recombination; OXPHOS: oxidative phosphorylation; PINK1: PTEN-induced kinase 1; ROS: reactive oxygen species; TCA: tricarboxylic acid; YY1: yin yang 1.

**Figure 5 cells-12-01237-f005:**
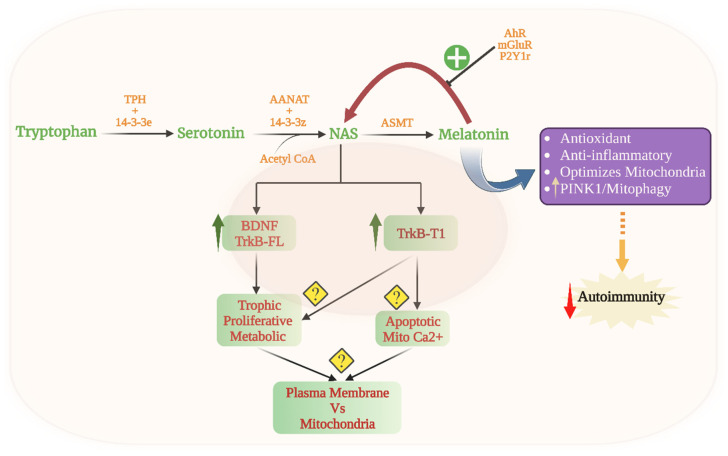
The figure shows the tryptophan-melatonin pathway (green shade). Tryptophan is metabolized by tryptophan hydroxylase (TPH). TPH requires stabilization by 14-3-3e to convert tryptophan to serotonin, which is the necessary precursor for the initiation of the melatonergic pathway. Serotonin can also be provided by neuronal inputs and other cellular sources, especially platelets. In the presence of acetyl-CoA, serotonin is converted by 14-3-3-stabilized AANAT to N-acetylserotonin (NAS), which is then converted by AANAT to melatonin. Aryl hydrocarbon receptor (AhR) activation ‘backward’ converts melatonin to NAS via O-demethylation as well as increasing the NAS/melatonin ratio, as can purinergic P2Y1r activation and glutamate activation of mGluR5. NAS can increase BDNF levels as well as activating TrkB receptors, both full-length (TrkB-FL) and truncated (TrkB-T1). TrkB-FL generally has trophic, proliferative and metabolic-enhancing effects, whilst TrkB-T1 has differential effects in different cells. The presence of TrkB on the mitochondrial membrane and plasma membrane may underlie TrkB-T1 differential actions in different cells. Melatonin has many protective effects as well as suppressing oxidative stress and MHC-1-linked autoimmunity. Abbreviations: AANAT: aralkylamine N-acetyltransferase; AhR: aryl hydrocarbon receptor; ASMT: N-acetylserotonin O-methyltransferase; BDNF: brain-derived neurotrophic factor; CYP: cytochrome P450; mGluR: metabotropic glutamate receptor; MHC-I: major histocompatibility complex-class I; Mito: mitochondria; NAS: N-acetylserotonin; P2Y1r: purinergic P2Y1 receptor; PINK1: PTEN-induced kinase 1; TrkB-FL: tyrosine receptor kinase B-full length; TrkB-T1: tyrosine receptor kinase B-truncated.

**Figure 6 cells-12-01237-f006:**
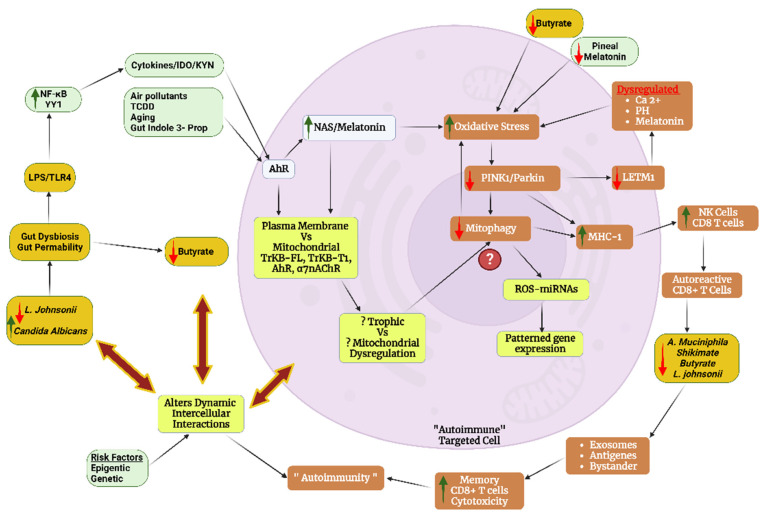
The figure shows how gut dysbiosis, gut permeability, pro-inflammatory cytokines, *Candida albicans* fungal infection and decreased pineal melatonin act to suppress melatonin production, upregulate the NAS/melatonin ratio and target the AhR in autoimmune targeted cells. The AhR ligands, including gut indole-3-propionate and derivatives, and NAS have differential effects at the mitochondrial AhR, TrkB-FL and TrkB-T1, as will acetylcholine at the mitochondrial a7nAChR (not shown for clarity). Gut dysbiosis/permeability and *Candida albicans* increase NF-kB and YY1 pro-inflammatory transcription factors leading to increased IDO and kynurenine activation of the AhR, along with many other AhR ligands. Gut dysregulation and suppressed pineal melatonin will also alter the homeostatic interactions within a given cell’s microenvironment, leading to intercellular changes that further challenge an ‘autoimmune’ susceptible cell. The suppression of the mitochondrial melatonergic pathway enhances oxidative stress, thereby decreasing PINK1 and its interactions with parkin and LETM1 on the mitochondrial membrane. Decreased PINK1 suppresses mitophagy, coupled with increased MHC-1 that drives ‘autoimmune’ processes via NK cell and CD8^+^ T cell attraction. Decreased PINK1 attenuates LETM1 phosphorylation, leading to Ca^2+^ and pH dysregulation, likely accompanied by alterations in how LETM1 interacts with 14-3-3 and/or AANAT in the regulation of the mitochondrial melatonergic pathway. Heightened oxidative stress, decreased mitophagy and suppressed melatonin increase ROS-driven miRNAs to change the patterned gene expression, relevant to both ‘autoimmune’ processes as well as dynamic intercellular interactions. Increased MHC-1 chemoattracts CD8^+^ T cells, with autoreactive CD8^+^ T cells being regulated by the gut via exosomes, antigens and/or bystander activation. The increased cytotoxicity of memory CD8^+^ T cells drives ‘autoimmune’ processes. Reduced *Akkermansia muciniphila* and limited shikimate pathway activation not only suppress tryptophan availability but also contribute to gut dysbiosis/permeability, thereby influencing the gut’s regulation of memory CD8^+^ T cells. Abbreviations: a7nAChR: alpha 7 nicotinic acetylcholine receptor; AhR: aryl hydrocarbon receptor; BDNF: brain-derived neurotrophic factor; IDO: indoleamine 2,3-dioxygenase; Indole-3-prop: indole-3-propionate; LETM1: leucine zipper-EF hand-containing transmembrane protein 1; LPS: lipopolysaccharide; MHC-1: major histocompatibility complex-class 1; NAS: N-acetylserotonin; NF-kB: nuclear factor kappa-light-chain-enhancer of activated B-cells; NK: natural killer; TCDD: 2,3,7,8-tetrachlorodibenzo-p-dioxin; TrkB-FL: tyrosine kinase receptor B-full length; TrkB-T1: tyrosine kinase receptor B-truncated; YY1: yin yang 1.

**Figure 7 cells-12-01237-f007:**
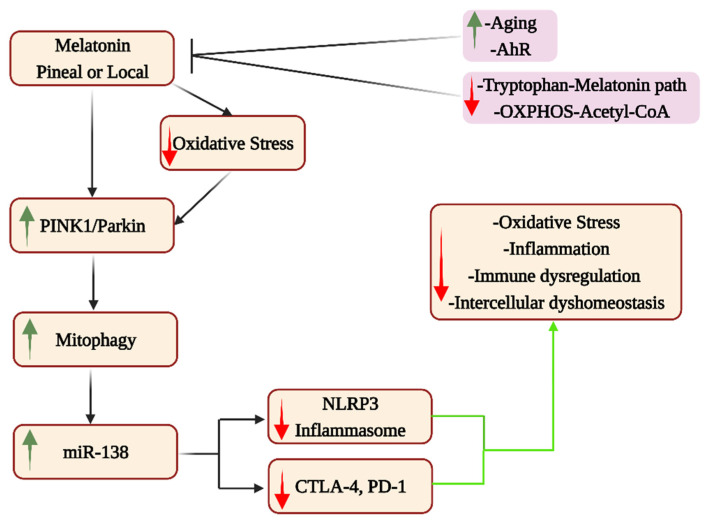
In conditions of suppressed mitophagy, melatonin (either pineal or local) suppresses oxidative stress to increase PINK1/parkin-induced mitophagy. Increasing mitophagy elevates miR-138, leading to a decrease in the NLRP3 inflammasome and inflammation, as well as decreasing the immune checkpoint inhibitors, CTLA-4 and PD-1, thereby promoting a balanced patterned immune response. This contributes to intercellular homeostatic interactions within a given microenvironment. A number of factors can suppress pineal and local melatonin, including aging-linked changes and increased levels and activation of the AhR, whilst factors acting to suppress the tryptophan-melatonin pathway and OXPHOS/Acetyl-CoA will inhibit pineal and local melatonin production. Abbreviations: AhR: aryl hydrocarbon receptor; CTLA-4: cytotoxic T-lymphocyte-associated molecule 4; miR: microRNA; NLRP3: NLR family pyrin domain containing 3; PD-1: programmed cell death-1; PINK1: PTEN-induced kinase 1.

**Figure 8 cells-12-01237-f008:**
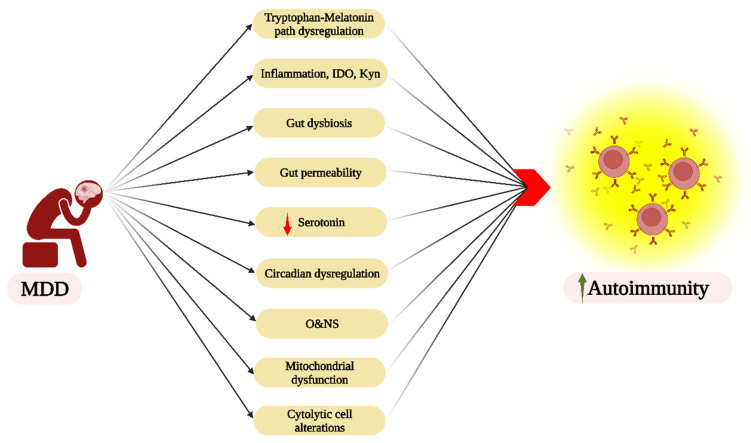
Shows how the pathophysiological underpinnings of MDD significantly overlap with those of ‘autoimmune’/‘immune-mediated’ disorders. Abbreviations: IDO: indoleamine, 2,3-dioxygenase; Kyn: kynurenine; MDD: major depressive disorder; O&NS: oxidative and nitrosative stress.

## Data Availability

Not applicable.

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
