# Peer review of "Redefining Autoimmune Disorders’ Pathoetiology: Implications for Mood and Psychotic Disorders’ Association with Neurodegenerative and Classical Autoimmune Disorders"

_cells, 2023, doi:10.3390/cells12091237_

Round 1
Reviewer 1 Report
I should note the importance of this present revision study. The linking among mitochondria-melatonin-gut microbiome toghether with the concept of those autoinmune processes that may occur on classical and complex pathologies is really interestingly and revelant. However, the work should be better organized to improve its comprehension for the reader. For instance, points 1,2 and 3 could be shorter. Point 4 should be the first. Additionally, some tables may help to summary that a lot data provided in text.
Author Response
Manuscript ID cells-2249765
Title Mitochondrial melatonergic pathway: Role in regulating autoimmune processes across diverse medical conditions, including Alzheimer’s disease, Parkinson’s disease, multiple sclerosis, cancer, type 1 diabetes mellitus, and neuropsychiatric disorders.
New Title: Redefining autoimmune disorders pathoetiology: implications for mood and psychotic disorders association with neurodegenerative and classical autoimmune disorders.
Authors George Anderson, Abbas F Almulla, Michael Maes, Russel J Reiter
Reviewer 1
I should note the importance of this present revision study. The linking among mitochondria-melatonin-gut microbiome toghether with the concept of those autoinmune processes that may occur on classical and complex pathologies is really interestingly and revelant. However, the work should be better organized to improve its comprehension for the reader. For instance, points 1,2 and 3 could be shorter. Point 4 should be the first. Additionally, some tables may help to summary that a lot data provided in text.
Response to Reviewer 1:
Thank you for these encouraging comments. Sections 1,2, and 3 have now been shortened. Section 4 now follows a general Introduction, as suggested. Six additional figures have now been added, and the two initial figures improved, which should help the reader to integrate the wide bodies of data mentioned.

Reviewer 2 Report
Thank you for the opportunity to review this manuscript, dealing with interesting findings entitled “Mitochondrial melatonergic pathway: Role in regulating autoimmune processes across diverse medical conditions, including Alzheimer’s disease, Parkinson’s disease, multiple sclerosis, cancer, type 1 diabetes mellitus, and neuropsychiatric disorders.” In this manuscript, they collected evidence and described well that the presence of the mitochondrial melatonergic pathways is crucial in driving CD8+ T cell and B-cell. Melatonin suppression, coupled with the upregulation of oxidative stress suppresses PINK1-driven mitophagy, raising levels of MHC-1, which underpins the chemoattraction of CD8+ T cells and the activation of antibody-producing B-cells. They proposed that ‘autoimmune’ disorders may conceptualize as ‘mitochondrial’ disorders, with mitochondrial melatonergic pathways. Overall, the review is strong in terms of writing, but it would be difficult to understand for the general readers due to the lack of sufficient figures, tables, and complex manuscript titles.
Since the author mentions all these medical conditions in the title the author needs to add a separate figure to show the molecular mechanism behind the Mitochondrial melatonergic pathway in terms of regulating autoimmune processes in all the medical conditions stated in the title (i.e., Alzheimer’s disease, Parkinson’s disease, multiple sclerosis, cancer, type 1 diabetes mellitus, and neuropsychiatric disorders.) or they may propose the molecular mechanism in relevant with all these medical conditions separately in one to two figures.
The author needs to present one or two tables with the previous finding stating “Mitochondrial melatonergic pathway in regulating autoimmune processes in different medical conditions”. It should cover in vitro, in vivo, and clinical observations of different medical conditions.
Some descriptions are missing such as Melatonin ameliorates excessive PINK1 mitophagy by enhancing SIRT1 expression that has a potential role in autoimmunity and how Melatonin-PINK1 signaling rescues amyloid pathology and mitochondrial dysfunction. Furthermore, Pink1 is expressed in different cell types. Is there any potential significance of its expression concerning melatonergic pathways and different medical conditions?
Author Response
Response to Reviewers
Manuscript ID cells-2249765
Title Mitochondrial melatonergic pathway: Role in regulating autoimmune processes across diverse medical conditions, including Alzheimer’s disease, Parkinson’s disease, multiple sclerosis, cancer, type 1 diabetes mellitus, and neuropsychiatric disorders.
New Title: Redefining autoimmune disorders pathoetiology: implications for mood and psychotic disorders association with neurodegenerative and classical autoimmune disorders.
Authors George Anderson, Abbas F Almulla, Michael Maes, Russel J Reiter
Reviewer 2:
Thank you for the opportunity to review this manuscript, dealing with interesting findings entitled “Mitochondrial melatonergic pathway: Role in regulating autoimmune processes across diverse medical conditions, including Alzheimer’s disease, Parkinson’s disease, multiple sclerosis, cancer, type 1 diabetes mellitus, and neuropsychiatric disorders.” In this manuscript, they collected evidence and described well that the presence of the mitochondrial melatonergic pathways is crucial in driving CD8+ T cell and B-cell. Melatonin suppression, coupled with the upregulation of oxidative stress suppresses PINK1-driven mitophagy, raising levels of MHC-1, which underpins the chemoattraction of CD8+ T cells and the activation of antibody-producing B-cells. They proposed that ‘autoimmune’ disorders may conceptualize as ‘mitochondrial’ disorders, with mitochondrial melatonergic pathways.
Response to Reviewer 2:
Thank you for these encouraging comments.
Reviewer 2:
Overall, the review is strong in terms of writing, but it would be difficult to understand for the general readers due to the lack of sufficient figures, tables, and complex manuscript titles.
Response to Reviewer 2:
We agree. Six additional figures have now been added to give a total of 8 figures, which should help the reader better integrate the wide bodies of data covered in this manuscript.
The title has also been changed and now reads:
“Redefining autoimmune disorders pathoetiology: implications for mood and psychotic disorders association with neurodegenerative and classical autoimmune disorders.”
Reviewer 2:
Since the author mentions all these medical conditions in the title the author needs to add a separate figure to show the molecular mechanism behind the Mitochondrial melatonergic pathway in terms of regulating autoimmune processes in all the medical conditions stated in the title (i.e., Alzheimer’s disease, Parkinson’s disease, multiple sclerosis, cancer, type 1 diabetes mellitus, and neuropsychiatric disorders.) or they may propose the molecular mechanism in relevant with all these medical conditions separately in one to two figures.
Response to Reviewer 2:
We agree. The title has now been changed to better accommodate the themed edition to which it has been submitted and now reads:
“Redefining autoimmune disorders pathoetiology: implications for mood and psychotic disorders association with neurodegenerative and classical autoimmune disorders.”
The emphasis of the manuscript is on core processes relevant across a diverse array of medical conditions, and following reviewer/editorial comments is now more integrated into the themed edition to which it has been submitted. It is highly likely that the processes driving this will be differentially modulated in different cell types. However, the detailing of this will require future data on core mitochondrial processes in specific cells across different medical conditions.
Reviewer 2:
The author needs to present one or two tables with the previous finding stating “Mitochondrial melatonergic pathway in regulating autoimmune processes in different medical conditions”. It should cover in vitro, in vivo, and clinical observations of different medical conditions.
Response to Reviewer 2:
We have now added 6 figures, giving a total of 8 figures, which highlight ‘core’ processes across medical conditions. Consequently, a table detailing melatonin effects would seem redundant, especially given the invariably beneficial effects of melatonin across medical conditions.
Reviewer 2:
Some descriptions are missing such as Melatonin ameliorates excessive PINK1 mitophagy by enhancing SIRT1 expression that has a potential role in autoimmunity and how Melatonin-PINK1 signaling rescues amyloid pathology and mitochondrial dysfunction. Furthermore, Pink1 is expressed in different cell types. Is there any potential significance of its expression concerning melatonergic pathways and different medical conditions?
Response to Reviewer 2:
Melatonin is well known as a homeostatic regulator (killing cancers but enhancing survival in non-neoplastic cells), with similar differential effects on excessive or suppressed autophagy. Melatonin increases PINK1/parkin driven autophagy in conditions where mitophagy is suppressed, which is typical of the medical conditions highlighted in this manuscript. The suppression of the tryptophan-melatonin pathway not only prevents endogenous melatonin from upregulating PINK1/parkin-driven mitophagy, but also desynchronizes NF-kB and YY1-induced BACE1/amyloid-B from melatonin production. The loss of melatonin production allows the ongoing inflammation/LPS/herpes virus to activate TLRs, leading NF-kB and YY1 induced amyloid-B in the absence of synchronized autocrine and paracrine melatonin, thereby preventing the negative feedback that melatonin would have on ongoing local inflammation. This contributes to local intercellular dyshomeostasis, as well as the toxic effects of excessive amyloid-B. The role of melatonin and gut microbiome-derived butyrate in the upregulation of sirtuin-1/sirtuin-3 is highlighted.
Round 2
Reviewer 2 Report
The revised manuscript and the author's response are satisfactory therefore the manuscript entitled "Redefining autoimmune disorders pathoetiology: implications for mood and psychotic disorders association with neurodegenerative and classical autoimmune disorders." could be published in the journal "Cells" in its present form.